# Sparse Autoencoders Learn Monosemantic Features in Vision-Language Models

**Mateusz Pach**[1,2,3,4]    **Shyamgopal Karthik**[1,2,3,4,5]    **Quentin Bouniot**[1,2,3,4]
**Serge Belongie**[6]    **Zeynep Akata**[1,2,3,4]

[1]Technical University of Munich    [2]Helmholtz Munich    [3]Munich Center for Machine Learning
[4]Munich Data Science Institute    [5]University of Tübingen    [6]University of Copenhagen
`mateusz.pach@tum.de`

## Abstract

Sparse Autoencoders (SAEs) have recently gained attention as a means to improve the interpretability and steerability of Large Language Models (LLMs), both of which are essential for AI safety. In this work, we extend the application of SAEs to Vision-Language Models (VLMs), such as CLIP, and introduce a comprehensive framework for evaluating monosemanticity at the neuron-level in visual representations. To ensure that our evaluation aligns with human perception, we propose a benchmark derived from a large-scale user study. Our experimental results reveal that SAEs trained on VLMs significantly enhance the monosemanticity of individual neurons, with sparsity and wide latents being the most influential factors. Further, we demonstrate that applying SAE interventions on CLIP's vision encoder directly steers multimodal LLM outputs (e.g., LLaVA), without any modifications to the underlying language model. These findings emphasize the practicality and efficacy of SAEs as an unsupervised tool for enhancing both interpretability and control of VLMs. Code and benchmark data are available at `https://github.com/ExplainableML/sae-for-vlm`.

## 1 Introduction

In recent years, Vision-Language Models (VLMs) like CLIP [47] and SigLIP [61] have gained widespread adoption, owing to their capacity for simultaneous reasoning over visual and textual modalities. They have found a surge of applications in various modalities, such as in audio [14, 57] and medicine [62], transferring to new tasks with minimal supervision. Yet our current understanding of VLM internals remains limited, necessitating methods that can systematically probe their representations. Sparse AutoEncoders (SAEs) [37] are an effective approach to probing the internal representations of such models. They efficiently discover concepts (abstract features shared between data points) through their simple architecture learned as a post-hoc reconstruction task. Although analysis with SAEs is popular for Large Language Models (LLMs) [7, 23, 48], for VLMs it has been limited to interpretable classification [35, 49], or the discovery of concepts shared across models [53].

Intuitively, SAEs reconstruct activations via a higher-dimensional space to disentangle distinct concepts from their overlapping representations in neural activations [7]. Neurons at different layers within deep neural networks are known to be naturally *polysemantic* [41], meaning that they can be strongly activated for multiple unrelated concepts such as cellphones and rulers. One common explanation for this behavior is the *superposition hypothesis* [3, 17], stating that concepts are encoded as linear combination of neurons. SAEs explicitly attempt to solve this issue by separating the entangled concepts into distinct representations. Despite their widespread use in research, the absence of a metric to evaluate SAEs at the *neuron-level* still hinders their practicality as an interpretation tool.

39th Conference on Neural Information Processing Systems (NeurIPS 2025).

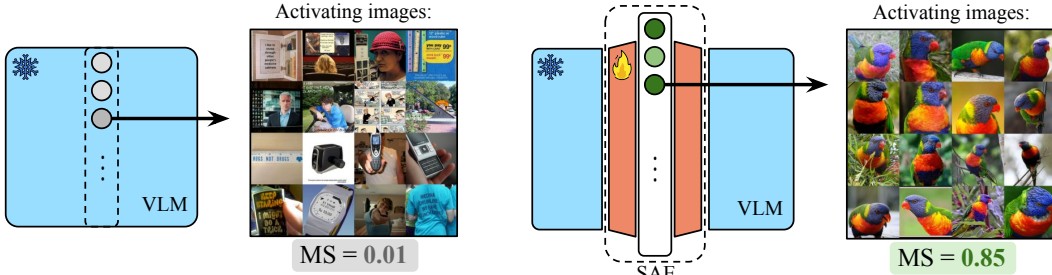

Figure 1: Sparse Autoencoder (SAE) in VLM (e.g. CLIP): Top activating images of a neuron in a pretrained VLM layer are polysemantic (left), and those of a neuron in a sparse latent of SAE trained to reconstruct the same layer are monosemantic (right), according to MonoSemanticity score (MS).

Discovering neurons encoding human-interpretable concepts requires analyzing them individually, which is even more tedious as layers of both the model and SAE are wide.

In this work, we quantitatively evaluate SAEs for VLMs through *monosemanticity*, defined as the similarity between inputs that strongly activate a neuron. We propose the MonoSemanticity score (MS) for vision tasks, that measures the pairwise similarity of images weighted by the activations for a given neuron. Unlike natural language where individual words require surrounding context (such as complete sentences) to clearly identify their meanings, individual images can directly activate neurons without additional context. We define highly activating images as images that strongly fire a particular neuron. Greater similarity among these images suggests that the neuron is more narrowly focused on a single concept, reflecting higher neuron monosemanticity. Using our MS score, we observe and validate that neurons in SAE are significantly more monosemantic (see Figure 1, right) than the original neurons (see Figure 1, left). The neuron of the original VLM typically has a low MS score, it fires for a wide range of objects, from *cellphones* to *rulers*. On the other hand, neurons within the SAE are more focused on a single concept, e.g. *parrots*, obtaining a higher score. This holds even for SAE with the same width as the original layer, implying that the sparse reconstruction objective inherently improves the separability of concepts. We further conduct a large-scale study to quantitatively assess alignment of our proposed MS score with human interpretation of monosemanticity. The results confirm that the difference between scores of two neurons strongly correlates with humans perceiving the higher-scoring neuron as more focused on a single concept.

Finally, we illustrate applicability of the monosemanticity of vision SAEs by transferring a CLIP-based SAE onto Multimodal LLMs (MLLMs), e.g. LLaVA [36]. Intervening on a single monosemantic SAE neuron in the vision encoder while keeping the LLM untouched allows steering the overall MLLM generated output to either *insert* or *suppress* the concept encoded in the selected SAE neuron. We summarize our contributions as follows:

- We propose the *MonoSemanticity score* (MS) for SAEs in vision tasks, that computes neuron-wise activation-weighted pairwise similarity of image embeddings. To validate our MS score against human judgment, we conduct a large-scale user study, the results of which can also serve as a benchmark for future research.

- We quantitatively compare MS between SAEs, and across their neurons. We find that Matryoshka SAE [9, 39] achieves overall superior MS, and that wider and sparser latents lead to better scores.

- We leverage the well-separability of concepts in SAE layers to intervene on neuron activations and steer outputs of MLLMs to *insert* or *suppress* any discovered concept.

## 2 Related Work

**Sparse Autoencoders.** Recent studies have repurposed traditional dictionary learning to enhance LLM and VLM interpretability [6, 46]. Specifically, there has been success in interpreting and steering LLMs with features learned by SAEs [16, 52]. Several enhancements to SAE mechanisms have been introduced, including new activation functions such as Batch TopK [8] or JumpReLU [48],

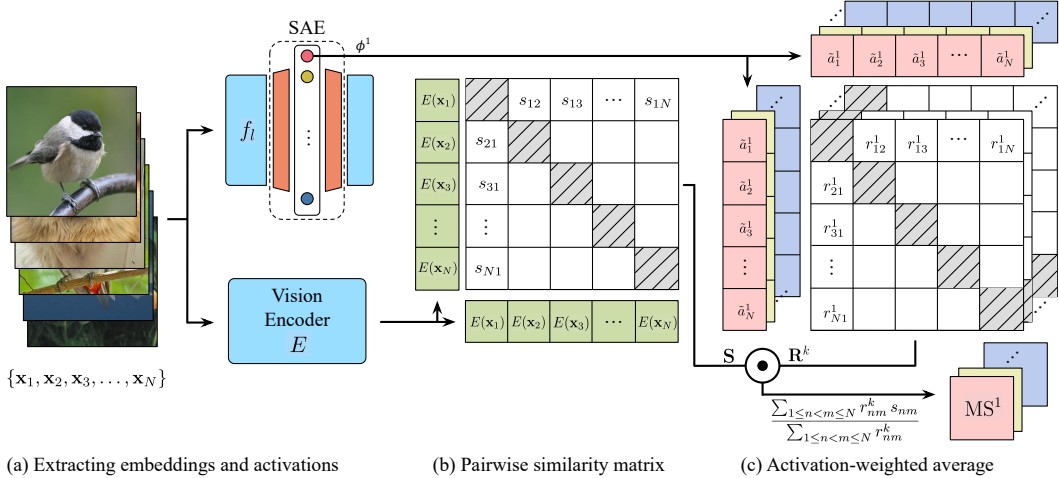

(a) Extracting embeddings and activations     (b) Pairwise similarity matrix     (c) Activation-weighted average

Figure 2: Computation of our MonoSemanticity score (MS). (a) Embeddings and activations are extracted for a set of images (b) to compute the pairwise embedding similarities and pairwise neuron activations. (c) MS is the average of embedding similarities weighted by the neuron activations.

and insights from Matryoshka representation learning [34] for SAEs [9, 39]. We analyze and evaluate neuron-level monosemanticity of SAEs in VLMs and their downstream uses.

**Vision-Language Models.** Since Contrastive Language–Image Pretraining (CLIP) [38, 47, 61], many models have emerged that align images and text in a shared embedding space [32, 56] or generate text conditioned on image inputs [12, 19, 36]. They have achieved strong results on benchmarks [59] and found many use-cases [5, 24, 26, 28]. As trust in these models has become a concern [4, 27, 60], understanding decision-making and ensuring safety, for example through steering, is increasingly important [2, 29]. Consequently, prior work has examined their internal representations [21, 22, 40, 45], uncovering interpretable neurons [15]. We demonstrate that SAEs enable more effective interpretation and control than directly operating on raw features.

**SAEs for VLMs.** Building on the success of SAEs in interpreting LLMs, researchers have tried applying them to vision and vision-language models, typically on CLIP [20, 35, 49, 50] and other vision encoders (e.g. DINOv2 [44]). There has also been interest in interpreting the denoising diffusion models [25] using SAEs [11, 30, 31, 51], discovering common concepts across different vision encoders [53], and applying them on multimodal LLMs [63]. Concurrently, monosemanticity of multimodal models has been investigated [58], although focusing on inter-modality differences. Our MS score provides a grounded, quantitative measure of the monosemanticity of individual neurons which is empirically aligned with human perception. Overall, we create a rigorous evaluation framework for SAEs in VLMs, as well as demonstrating their use for steering multimodal LLMs.

## 3 Sparse Autoencoders for VLMs

### 3.1 Background and Formulation of SAEs

SAEs implement a form of sparse dictionary learning, where the goal is to learn a sparse decomposition of a signal into an overcomplete dictionary of atoms [42]. More specifically, an SAE consists of linear layers $\mathbf{W}_{\text{enc}} \in \mathbb{R}^{d \times \omega}$ and $\mathbf{W}_{\text{dec}} \in \mathbb{R}^{\omega \times d}$ as *encoder* and *decoder*, with a non-linear activation function $\sigma : \mathbb{R}^{\omega} \to \mathbb{R}^{\omega}$. Both layers share a bias term $\mathbf{b} \in \mathbb{R}^{d}$ subtracted from the encoder's input and later added to the decoder's output. The *width* $\omega$ of the latent SAE layer is chosen as a factor of the original dimension, such that $\omega := d \times \varepsilon$, where $\varepsilon$ is called the *expansion factor*.

In general, SAEs are applied on embeddings $\mathbf{v} \in \mathbb{R}^{d}$ of a given layer $l$ of the model to explain $f : \mathbb{X} \to \mathbb{Y}$, such that $f_l : \mathbb{X} \to \mathbb{R}^{d}$ represents the composition of the first $l$ layers and $\mathbb{X} \subset \mathbb{R}^{d_i}$ is the space of input images. A given input image $\mathbf{x} \in \mathbb{X}$ is first transformed into the corresponding embedding vector $\mathbf{v} := f_l(\mathbf{x})$, before being decomposed by the SAE into a vector of activations $\phi(\mathbf{v}) \in \mathbb{R}^{\omega}$, and its reconstruction vector $\hat{\mathbf{v}} \in \mathbb{R}^{d}$ is obtained by:

$$\phi(\mathbf{v}) := \sigma(\mathbf{W}_{\text{enc}}^\top(\mathbf{v} - \mathbf{b})), \quad \psi(\mathbf{v}) := \mathbf{W}_{\text{dec}}^\top\mathbf{v} + \mathbf{b}, \quad \hat{\mathbf{v}} := \psi(\phi(\mathbf{v})). \tag{1}$$

The linear layers $\mathbf{W}_{\text{enc}}$ and $\mathbf{W}_{\text{dec}}$ composing the SAE are learned through a reconstruction objective $\mathcal{R}$ and sparsity regularization $\mathcal{S}$, to minimize the following loss:

$$\mathcal{L}(\mathbf{v}) := \mathcal{R}(\mathbf{v}) + \lambda\mathcal{S}(\mathbf{v}), \tag{2}$$

where $\lambda$ is a hyperparameter governing the overall sparsity of the decomposition. The most simple instantiation [7, 49] uses a ReLU activation, an $L^2$ reconstruction objective and an $L^1$ sparsity penalty, such that

$$\sigma(\cdot) := \text{ReLU}(\cdot), \quad \mathcal{R}(\mathbf{v}) := \|\mathbf{v} - \hat{\mathbf{v}}\|_2^2, \quad \mathcal{S}(\mathbf{v}) := \|\phi(\mathbf{v})\|_1. \tag{3}$$

The (Batch) TopK SAEs [8, 23, 37] use a TopK activation function governing the sparsity directly through $K$. Finally, Matryoshka SAEs [9, 39] group neuron activations $\phi^i(\mathbf{v})$ into different levels of sizes $\mathcal{M}$, to obtain a nested dictionary trained with multiple reconstruction objectives:

$$\mathcal{R}(\mathbf{v}) := \sum_{m \in \mathcal{M}} \|\mathbf{v} - \mathbf{W}_{\text{dec}}^\top\phi^{1:m}(\mathbf{v})\|_2, \tag{4}$$

where $\phi^{1:m}$ corresponds to keeping only the first $m$ neuron activations, and setting the others to zero. It is important to note that Matryoshka SAEs can be combined with any SAE variant, e.g. with BatchTopK [9] or ReLU [39], as only the reconstruction objective is modified.

## 3.2 Monosemanticity Score

A neuron's interpretability increases as its representation becomes disentangled into a single, clear concept. Therefore, quantifying the monosemanticity of individual neurons helps identify the most interpretable ones, while aggregating these scores across an entire layer allows assessing the overall semantic clarity and quality of the representations learned by the SAE. We propose measuring monosemanticity by computing pairwise similarities between images that strongly activate a given neuron, where high similarity indicates these images likely represent the same concept. These similarities can be efficiently approximated using deep embeddings from a pretrained image encoder $E$. Since selecting a fixed number of top-activating images is challenging due to varying levels of specialization across neurons, we instead evaluate monosemanticity over a large, diverse set of unseen images, weighting each image by its activation strength for the neuron.

We formally describe our proposed MonoSemanticity score (MS) below, with an illustration given in Figure 2. This metric can be computed for each of the $\omega$ neurons extracted from the SAE. Given a diverse set of images $\mathcal{I} = \{\mathbf{x}_n \in \mathbb{X}\}_{n=1}^N$, and a pretrained image encoder $E$, we first extract embeddings to obtain a pairwise similarity matrix $\mathbf{S} = [s_{nm}]_{n,m} \in [-1,1]^{N \times N}$, which captures semantic similarity between each pair of images. The similarity $s_{nm}$ of the pair $(\mathbf{x}_n, \mathbf{x}_m)$ is computed as the cosine similarity between the corresponding pair of embedding vectors:

$$s_{nm} := \frac{E(\mathbf{x}_n) \cdot E(\mathbf{x}_m)}{|E(\mathbf{x}_n)||E(\mathbf{x}_m)|}. \tag{5}$$

We then collect activation vectors $\{\mathbf{a}^k = [a_n^k]_n \in \mathbb{R}^N\}_{k=1}^\omega$ across all $\omega$ neurons, for all images in the dataset $\mathcal{I}$. Specifically, for each image $\mathbf{x}_n$, the activation of the $k$-th neuron:

$$\mathbf{v}_n := f_l(\mathbf{x}_n), \quad a_n^k := \phi^k(\mathbf{v}_n), \tag{6}$$

where $l$ represents the layer at which the SAE is applied, $f_l$ is the composition of the first $l$ layers of the explained model, and $\phi^k$ is the $k$-th neuron of $\phi(\mathbf{v}_n)$ (or of $\mathbf{v}_n$ when evaluating neurons of the original layer $l$ of $f$). To ensure a consistent activation scale, we apply min-max normalization to each $\mathbf{a}^k$, yielding $\tilde{\mathbf{a}}^k := [\tilde{a}_n^k]_n \in [0,1]^N$, where

$$\tilde{a}_n^k = \frac{a_n^k - \min_{n'} a_{n'}^k}{\max_{n'} a_{n'}^k - \min_{n'} a_{n'}^k}. \tag{7}$$

Using these normalized activations, we compute a relevance matrix $\mathbf{R}^k = [r_{nm}^k]_{n,m} \in [0,1]^{N \times N}$ for each one of the $\omega$ neurons, which quantifies the shared neuron activation of each image pair:

$$r_{nm}^k := \tilde{a}_n^k \tilde{a}_m^k. \tag{8}$$

Finally, our proposed score $\text{MS}^k \in [-1,1]$ for the $k$-th neuron is computed as the average pairwise similarity weighted by the relevance, without considering same image pairs $(\mathbf{x}_n, \mathbf{x}_n)$:

$$\text{MS}^k := \frac{\sum_{1 \le n < m \le N} r_{nm}^k s_{nm}}{\sum_{1 \le n < m \le N} r_{nm}^k} \tag{9}$$

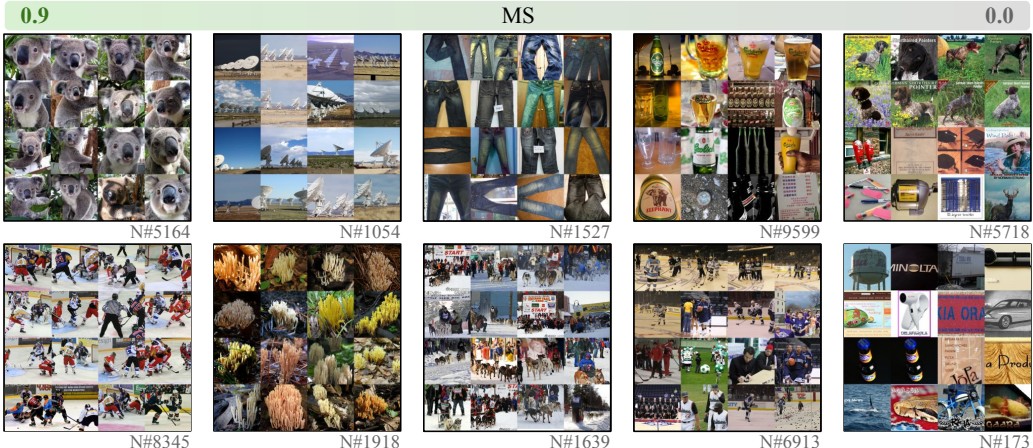

Figure 3: Top activating images of neurons with MonoSemanticity (MS) scores ranging from high (left) to low (right). Higher scores correlate with more similar images, reflecting monosemanticity.

### 3.3 Steering MLLMs with Vision SAEs

Finding monosemantic neurons is not only useful for interpretability. The SAEs let us induce controllable semantic biases in the response of MLLMs without modifying the underlying model parameters, and without touching any textual part. In other words, we can steer the model into *seeing or not* targeted concepts in the image. Our MS score becomes a strong tool to help select the most monosemantic neurons for a precise and efficient steering.

We first describe LLaVA [36] as an example MLLM architecture. The LLaVA model $g : \mathbb{X} \times \mathbb{T} \to \mathbb{T}$ expects a pair of image and text $(\mathbf{x}, \mathbf{t})$ and outputs a text answer $\mathbf{o}$, where $\mathbb{T} \subset \mathbb{R}^{d_t}$ is the word embedding space. Internally, it converts the image $\mathbf{x}$ into $t_{\mathbf{x}} \in \mathbb{N}$ token embeddings $\{\mathbf{v}_i\}_{i=1}^{t_{\mathbf{x}}}$ obtained from vision encoder $f_l : \mathbb{X} \to \mathbb{R}^{d \times t_{\mathbf{x}}}$ composed of the first $l$ layers of CLIP [47]. These embeddings are then projected into visual tokens $\mathbf{H}_{\mathbf{x}} \in \mathbb{R}^{d_t \times t_{\mathbf{x}}}$ in the word embedding space, and are finally fed along with tokenized text $\mathbf{H}_{\mathbf{t}} \in \mathbb{R}^{d_t \times t_{\mathbf{t}}}$ into the pretrained LLM (e.g. LLaMA [54] or Vicuna [10]) to obtain the output text $\mathbf{o}$.

We modify this architecture by injecting a pretrained SAE $(\phi, \psi)$ of width $\omega$ at the *token-level* after the vision encoder $f_l$. For all token embeddings $\mathbf{v}_i \in \mathbb{R}^d, i \in \{1, \dots, t_{\mathbf{x}}\}$, we first extract the SAE decomposition into activation $\mathbf{a}_i := \phi(\mathbf{v}_i) \in \mathbb{R}^{\omega}$ across all neurons. After identifying the neuron $k \in \{1, \dots, \omega\}$ representing the targeted concept, to steer the overall model $g$ towards this concept, we manipulate the SAE activations of *all token embeddings* for the neuron $k$ to obtain $\{\hat{\mathbf{a}}_i \in \mathbb{R}^{\omega}\}_{i=1}^{t_{\mathbf{x}}}$:

$$\forall j \in \{1, \dots, \omega\}, \quad \hat{a}_i^j = \begin{cases} \alpha, & j = k \\ a_i^j, & j \neq k \end{cases} \tag{10}$$

where $\alpha \in \mathbb{R}$ is the intervention value we want to apply to the activation of neuron $k$. Finally, we decode the manipulated activation vectors for each token $\hat{\mathbf{a}}_i$ back into a manipulated token embedding $\hat{\mathbf{v}}_i = \psi(\hat{\mathbf{a}}_i) \in \mathbb{R}^d$ with the SAE decoder. Token embeddings are then processed as usual to generate the steered LLaVA's response. We include an illustration of the overall process in the Appendix.

## 4 Experiments

### 4.1 Experimental Settings

We apply SAEs to explain fixed and pretrained CLIP ViT-L/14-336px [47], SigLIP SoViT-400m/14-384px [61], AIMv2 L/14-224px [19], and WebSSL MAE-300m/14-224px [18]. The SAEs are trained on activation vectors pre-extracted from the model's responses to ImageNet [13] images. For CLIP, activation vectors are extracted from the classification (CLS) tokens in the residual stream after layers $l \in \{11, 17, 22, 23\}$, or from the output of the final projection layer. For steering experiments, however, the SAEs are trained on activation vectors corresponding to two random token embeddings

per image, taken from layer $l = 22$. For other encoders, we similarly use the CLS tokens from the final layers, or two random token embeddings if a CLS token is not available.

In the following sections, we are interested in both BatchTopK [8] and Matryoshka Batch-TopK SAEs [9] variants. If not stated otherwise, we set the groups of Matryoshka SAEs as $\mathcal{M} = \{0.0625\omega, 0.1875\omega, 0.4375\omega, \omega\}$, which roughly corresponds to doubling the size of the number of neurons added with each level down. For the BatchTopK activation, we fix the maximum number of non-zero latent neurons to $K = 20$. Both SAE types are compared across a wide range of expansion factors $\varepsilon \in \{1, 2, 4, 8, 16, 64\}$. All SAEs are optimized for $10^5$ steps with minibatches of size 4096 using Adam optimizer [33], with the learning rate initialized at $\frac{16}{125\sqrt{\omega}}$ following previous work [23]. To measure SAE performance, we use $R^2$ for reconstruction quality and the $L^0$ norm for the activation sparsity of $\phi(\mathbf{v})$. Throughout the paper, we quantify MS of neurons using DINOv2 ViT-B [44] as image encoder $E$, and present more analysis with different encoders in Appendix. Experiments are run on a single NVIDIA A100 GPU.

### 4.2 Evaluating Interpretability of VLM Neurons

#### 4.2.1 Alignment of MS with human perception

We first illustrate the correlation between MS score and the underlying monosemanticity of neurons, with examples in Figure 3 of the 16 highest activating images for neurons with decreasing MS from left to right. We observe that the highest scoring neurons (with MS = 0.9, on the far left) are firing for images representing the same object, i.e. close-up pictures of a *koala* (on the top) and a *hockey* (on the bottom). As the score decreases, the corresponding neurons fire for less similar or even completely different objects or scenes. To verify this observation in a more quantitative way, we conducted a large scale user study on the Mechanical Turk platform. This

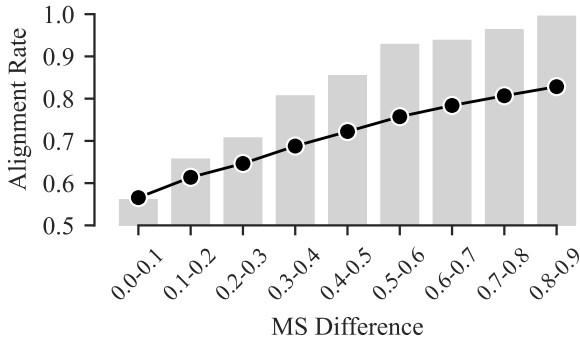

Figure 4: Alignment Rate (AR, %) of humans with MS score when judging which neuron in a pair is more monosemantic, grouped by MS difference between the neurons. Bars show AR per interval; dots show cumulative AR up to that interval.

study resulted in a total of 1000 questions across 71 unique users, with 3 answers per question aggregated through majority voting. Results of this study are presented in Figure 4, and we provide more details on the setup in Appendix. When asked to select the set of more monosemantic images from a pair of sets $(a, b)$, the users answered in accordance to the MS in $82.8\%$ of the cases, assuming $\delta = |\text{MS}(a) - \text{MS}(b)| \sim \mathcal{U}(0, 0.9)$. This alignment rate monotonically raises from $56.6\%$ for $\delta \in (0.0, 0.1)$ to $100.0\%$ for $\delta \in (0.8, 0.9)$, highlighting that users especially agree with MS as the difference in monosemanticity between the two sets becomes more pronounced. This demonstrates that MS can be used as a reliable measure aligning well with human perception of similarity. Detailed results of the user study are released for future benchmark of image similarity and monosemanticity.

#### 4.2.2 Monosemanticity of SAEs

In Table 1a, we report MS of the highest scoring neurons of two SAE types (BatchTopK [8] and Matryoshka BatchTopK [9]) trained at different layers with various expansion factors $\varepsilon$. We also include results for original neurons of the corresponding layer decomposed by SAEs ("No SAE"). We observe that SAEs' neurons consistently have significantly higher MS for their best neuron when compared to original ones, implying that SAEs are better separating and disentangling concepts between their neurons. Interestingly, while the highest MS score is increasing with higher expansion factor $\varepsilon$, i.e. with increased width $\omega$ of the SAE layer, this holds true already for expansion factor $\varepsilon = 1$, meaning that the disentanglement of concepts is also linked to the sparse dictionary learning and not only to the increased dimensionality. Finally, comparing SAE variants, we observe that while the Matryoshka reconstruction objective improves the concept separation at same expansion factor, it also achieves about 2 or 3 points lower $R^2$ for the same expansion factors (more details in Appendix).

Table 1: Sparse Autoencoders (SAEs) decompose "No SAE" neurons into more monosemantic units as shown by MonoSemanticity (MS) score. Higher SAE expansion factors yield higher MS scores.

(a) Highest MS scores of neurons in various CLIP ViT-Large [47] layers.

| SAE type | Layer | No SAE | Expansion factor | | | | | |
|---|---|---|---|---|---|---|---|---|
| | | | ×1 | ×2 | ×4 | ×8 | ×16 | ×64 |
| BatchTopK [8] | 11 | 0.01 | 0.61 | 0.73 | 0.71 | 0.87 | 0.90 | 1.00 |
| | 17 | 0.01 | 0.65 | 0.79 | 0.86 | 0.86 | 0.93 | 1.00 |
| | 22 | 0.01 | 0.66 | 0.79 | 0.80 | 0.88 | 0.92 | 1.00 |
| | 23 | 0.01 | 0.73 | 0.72 | 0.83 | 0.89 | 0.93 | 1.00 |
| | last | 0.01 | 0.57 | 0.78 | 0.78 | 0.81 | 0.85 | 1.00 |
| Matryoshka [9, 39] | 11 | 0.01 | 0.84 | 0.90 | 0.95 | 1.00 | 0.89 | 1.00 |
| | 17 | 0.01 | 0.86 | 0.84 | 0.93 | 0.94 | 0.96 | 1.00 |
| | 22 | 0.01 | 0.83 | 0.83 | 0.87 | 0.94 | 1.00 | 1.00 |
| | 23 | 0.01 | 0.82 | 0.84 | 0.89 | 0.93 | 0.96 | 1.00 |
| | last | 0.01 | 0.82 | 0.91 | 0.89 | 0.93 | 0.91 | 1.00 |

(b) Top MS scores of neurons from last layers of different vision encoders. Improvements in MS score from applying Matryoshka SAEs are consistent across all the models.

| Vision Encoder | No SAE | Exp. factor | |
|---|---|---|---|
| | | ×1 | ×4 |
| WebSSL [18] | 0.01 | 0.79 | 0.92 |
| CLIP [47] | 0.01 | 0.82 | 0.89 |
| SigLIP [61] | 0.01 | 0.83 | 0.88 |
| AIMv2 [19] | 0.01 | 0.59 | 0.85 |

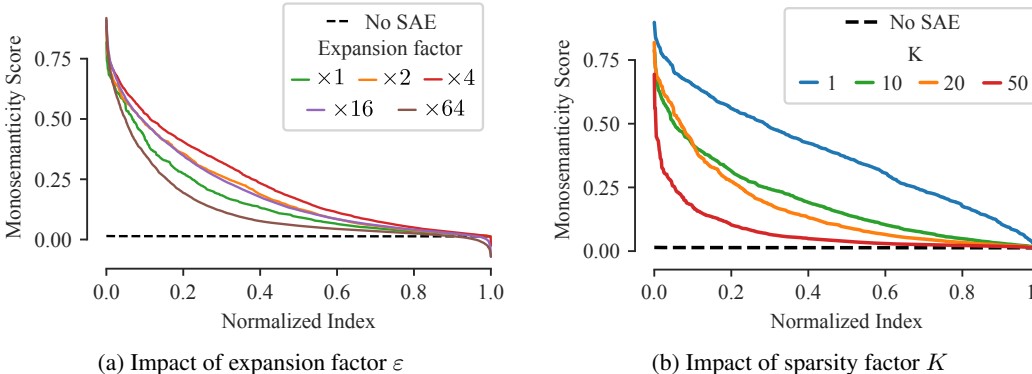

(a) Impact of expansion factor $\varepsilon$      (b) Impact of sparsity factor $K$

Figure 5: MonoSemanticity scores in decreasing order across neurons, normalized by width. Results are shown for the last layer of the model, without SAE ("No SAE", in black dashed line), and with SAE in straight lines using either (a) different expansion factors ($\varepsilon = 1$, $\varepsilon = 2$, $\varepsilon = 4$, $\varepsilon = 16$, $\varepsilon = 64$) or (b) different sparsity levels ($K = 1$, $K = 10$, $K = 20$, and $K = 50$).

Table 1b presents the highest observed MS scores among neurons in the last layers of various image encoders, both before ("No SAE") and after attaching Matryoshka SAEs. We find that the SAE latent neurons outperform the original neurons in every case. As before, increasing the expansion factor $\varepsilon$ helps discover more monosemantic units. This suggests the universality of the SAE approach across vision representations derived from different training objectives.

To analyze monosemanticity across all neurons, we plot in Figure 5a the scores of both the original neurons and the Matryoshka SAE neurons from the last layer of model $f$. We ordered neurons by decreasing scores and normalized neuron indices to the $[0, 1]$ interval to better compare SAEs with different widths. These results confirm our analysis above, and demonstrate that the vast majority of neurons within SAEs have improved MS compared to the original neurons. Even when comparing with $\varepsilon = 1$, i.e. with same width between the SAE and original layers, we can see that about $90\%$ of the neurons within the SAE have better scores than the original neurons, proving once again that the sparse decomposition objective inherently induces a better separation of concepts between neurons. Furthermore, MS scores increase overall with the expansion factor until a certain point ($\varepsilon = 4$), after which they decrease overall, reaching even lower values than $\varepsilon = 1$. Although the relative fraction of neurons at different values of MS is decreasing for very wide latents, the absolute number is still increasing. We refer the reader to Appendix for MS with raw (unnormalized) neuron indices.

The relationship between the sparsity level $K$ used when training Matryoshka SAEs and the scores of the learned neurons is illustrated in Figure 5b. We observe that a stricter sparsity constraint decomposes the representation into more monosemantic features overall. However, this does not

Table 2: Percentage of generations meeting evaluation criteria for concept insertion and suppression. SAE-derived steering directions yield higher success rates than Difference-in-Means (DiffMean) [2].

(a) Concept insertion

| | Ours | DiffMean |
|---|---|---|
| Desired concept appeared | 48.7 | **53.1** |
| Base prompt followed | **85.8** | 66.2 |
| **Both criteria satisfied** | **42.4** | 35.8 |

(b) Concept suppression

| | Ours | DiffMean |
|---|---|---|
| Desired concept removed | **64.4** | 64.0 |
| Unrelated concept kept | **81.4** | 38.7 |
| **Both criteria satisfied** | **52.5** | 33.3 |

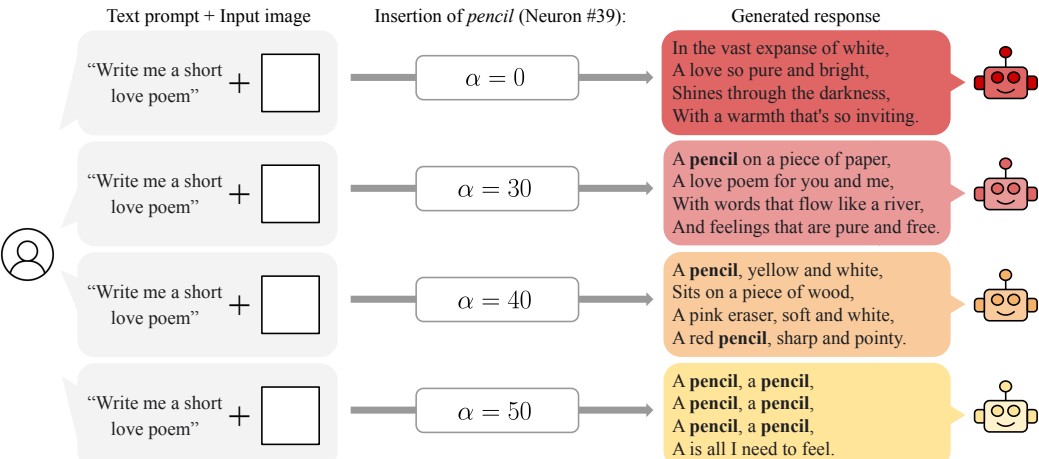

Figure 6: Steered LLaVA outputs after clamping the activation values of a chosen neuron, i.e. Neuron #39 of SAE in CLIP corresponding to *pencil*. Initially, the poem follows the given instruction (the prompt and white image), but as the intervention weight $\alpha$ increases, it becomes increasingly influenced by the neuron's concept, first mentioning the *pencil*'s attributes, then the *pencil* itself. This shows that our interventions enable new capabilities for the unsupervised steering of these models.

imply that the highest sparsity ($K = 1$) is always the best choice, as improvements in MS come at the cost of reduced reconstruction quality. In the same setup, the $R^2$ varies from $31.3\%$ at the lowest $K = 1$ to $74.9\%$ at the highest $K = 50$. To balance interpretability and reconstruction quality, we set $K = 20$ for which the $R^2$ remains at a reasonable $66.8\%$. Detailed results are in the Appendix.

### 4.3  Steering Multimodal LLMs

We train the Matryoshka BatchTopK SAE [9] with expansion factor $\varepsilon = 64$ on random token embeddings from layer $l = 22$ of the CLIP vision encoder obtained for the ImageNet training data. The trained SAE is plugged after the vision encoder of LLaVA-1.5-7b [36] (uses Vicuna [10] LLM).

**Quantitative Results.** We first compare the performance of SAE-based steering for VLMs against Difference-in-Means (DiffMean) [2], a popular approach based on activation steering. For each of 100 SAE neurons in LLaVA, we identify its top-activating image grid. To perform concept insertion, we boost the neuron's activation and prompt LLaVA with 10 diverse text queries, such as "Propose a math word problem," or "Invent a new holiday," then evaluate if the output contains the concept and still responds to the prompt. For concept supression, we apply a negative intervention, ask LLaVA to describe the images, and check whether the concept is removed and unrelated images are still described correctly. The evaluation is done with a LLM-as-a-judge setup using GPT-4.1-mini [43]. We provide the full details of the prompts and the evaluation procedure in Appendix B. The results in Table 2 demonstrate that SAE-based directions outperform DiffMean in both concept insertion and suppression. For insertion, SAE effectively introduces the intended concept ($48.7\%$ vs. $53.1\%$) while maintaining much stronger adherence to the base prompt ($85.5\%$ vs. $66.2\%$). For suppression, it performs on par with DiffMean in removing the target concept ($64.4\%$ vs. $64.0\%$), yet far surpasses it in preserving unrelated content ($81.4\%$ vs. $38.7\%$).

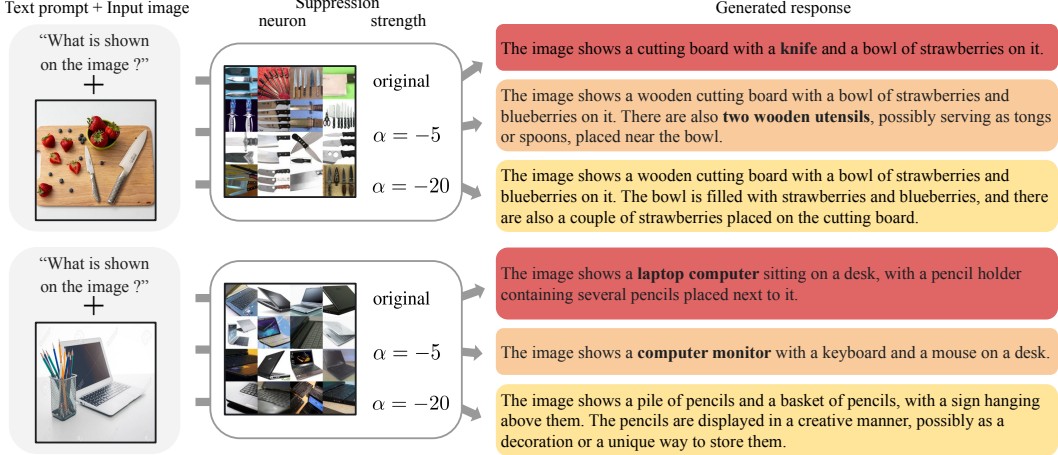

Figure 7: Overriding neuron activations with negative values allows concept suppressing. Originally, LLaVA describes the images by mentioning all the objects visible on the input images. However, as we decrease the value $\alpha$ of neurons associated with *knives* and *laptops*, it first confuses them with wooden utensils and computer monitor, then eventually ignores them completely. At the same time, it continues to faithfully describe other objects like wooden board, strawberries, and pencil holder.

To further confirm the steering capabilities of SAE neurons, we also perform a sanity check on the steering capabilities of the SAE neurons. We directly compare CLIP similarity scores between the top-16 images activating a given neuron and the corresponding text outputs, both before and after steering. We prompt the model with "What is shown on the image? Use exactly one word." and compare its original answer to the one generated after fixing the activation of a specific SAE neuron to $\alpha = 100$, varying one neuron at a time. In the first setup, a white image is used while intervening on the first 1000 neurons to isolate the neuron manipulation's effect. In the second, 1000 random ImageNet images are used while steering only 10 neurons to test effects on natural inputs. The results in Table 3 clearly illustrates that steering increases image-text similarity scores. For context, we compute reference similarities in CLIP space: the average similarity between each ImageNet class image and its class name sets an upper bound ($0.283 \pm 0.034$), while random image–class pairs set a lower bound ($0.185 \pm 0.028$). Neuron steering yields a relative gain of 22% within this range, highlighting the significance of the results.

Table 3: Mean similarity of neurons' activating images to output word, with and without steering, on white or random ImageNet images. Upper bound with correct image-classname pairs is $0.283$, lower bound with random pairs is $0.185$.

| Steering | White Image | ImageNet |
|---|---|---|
| ✓ | **$0.259 \pm 0.036$** | **$0.263 \pm 0.037$** |
| ✗ | $0.212 \pm 0.021$ | $0.211 \pm 0.028$ |

**Qualitative Examples.** Figure 6 illustrates the effectiveness of *concept insertion* by manipulating a single neuron of the SAE. We prompt the model with the instruction "Write me a short love poem", along with a white image. By intervening on an SAE neuron associated to the *pencil* concept and increasing the corresponding activation value, we observe the impact on the generated output text. While the initial output mentions the "white" color and focuses on the textual instruction, i.e. "love" poem, the output becomes more and more focused on *pencil* attributes as we manually increase the intervention value $\alpha$ (most highly activating images for the selected neuron is in Appendix) until it only mentions *pencils*. We provide more examples with a different input prompt ("Generate a scientific article title") in Appendix, for which steered LLaVA exhibits a similar behavior.

In Figure 7, we show that, by clamping a specific neuron to negative value, we can suppress a concept. LLaVA is asked to answer what is shown on the images given a photo of a cutting board with knives and strawberries and a photo of laptop and pencil holder full of pencils. By default the model generates correct descriptions containing all the objects. However, when we intervene by decreasing the activation value $\alpha$ of the neurons associated with *knife* and *laptop*, the resulting descriptions progressively omit these concepts. This provides a promising strategy for filtering out harmful or undesired content at an early stage, before it even reaches the language model.

# 5 Conclusion

We introduced the MonoSemanticity score (MS), a quantitative metric for evaluating monosemanticity at the neuron level in SAEs trained on VLMs. Our analysis revealed that SAEs primarily increased monosemanticity through sparsity and wider latents and highlighted the superior performance of Matryoshka SAEs. We further verified the alignment of MS with human perception through a large-scale human study. Leveraging the clear separation of concepts encoded in SAEs, we explored their effectiveness for unsupervised, concept-based steering of multimodal LLMs, highlighting a promising direction for future research. Potential extensions of this work include adapting our metric to text representations and investigating the interplay between specialized (low-level) and broad (high-level) concepts within learned representations.

**Limitations.** We focused our evaluation on various SAE architectures, the most common dictionary learning implementations that scale effectively to large VLMs. However, our MS metric is model-agnostic and could be applied to study other comparable approaches as well. High MS score neurons do not always produce precise effects when used to steer MLLM outputs. For example, a *golden retriever* neuron from an SAE trained on ImageNet can trigger any dog-related output. This could happen because, while SAEs can disentangle detailed classes in the dataset, MLLMs may have limited fine-grained understanding and may not be perfectly aligned with the vision encoder. Moreover, a fraction of the SAE neurons that act as feature detectors do not exhibit any clear steering effect [1].

## Acknowledgments and Disclosure of Funding

This work was partially funded by the ERC (853489 - DEXIM) and the Alfried Krupp von Bohlen und Halbach Foundation, which we thank for their generous support. We are also grateful for partial support from the Pioneer Centre for AI, DNRF grant number P1. Shyamgopal Karthik thanks the International Max Planck Research School for Intelligent Systems (IMPRS-IS) for support. Mateusz Pach would like to thank the European Laboratory for Learning and Intelligent Systems (ELLIS) PhD program for support.

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

# Contents

# A  Broader Impact

Our work contributes to the field of interpretability and alignment, which are essential components for building safe AI systems. Our MonoSemanticity score provides a new way to evaluate the effectiveness of recently popular dictionary learning methods, such as sparse autoencoders (SAEs), by incorporating human judgment into the evaluation process. This makes it easier to assess and build trust in systems that use SAEs.

In addition, we show that SAEs can be highly effective in steering applications. They can be used to encourage or discourage specific behaviors in models, or to help models recognize or ignore certain concepts, including potentially dangerous ones. This is especially useful for ensuring that models produce desired outputs and remain aligned with human values and goals.

# B  More details on steering

We illustrate in Figure A1 how we steer LLaVA-like models. We separately train SAEs on top of the pretrained CLIP vision encoder to reconstruct the *token embeddings* $\mathbf{v}_i$, and then attach it back after the vision encoder during inference. Intervening on a neuron within the SAE layer steers the reconstructed tokens $\hat{\mathbf{v}}_i$ towards the activated concept, which then steers the LLM's generated output. We present in Figure A2 additional examples of LLaVA prompted to generate scientific titles, and the outputs before and after intervening on SAE neurons. Increasing the activation of specific neurons will modify the outputs to include elements from images highly activating the corresponding neuron.

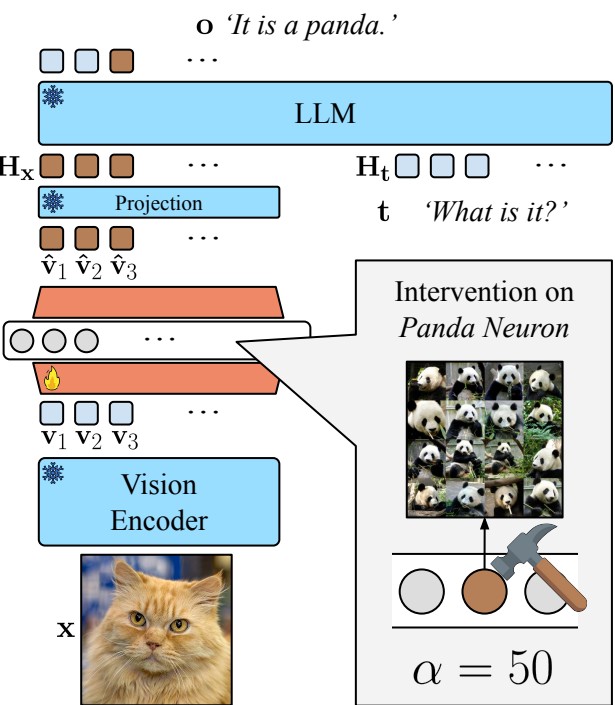

Figure A1: LLaVA-like models can be steered towards *seeing* a concept (e.g. *panda*) not present in the input image $\mathbf{x}$. By attaching SAE after vision encoder and intervening on its neuron representing that concept, we effectively manipulate the LLM's response. Such flexible and precise steering is possible thanks to the extensive concept dictionary identified through the SAE.

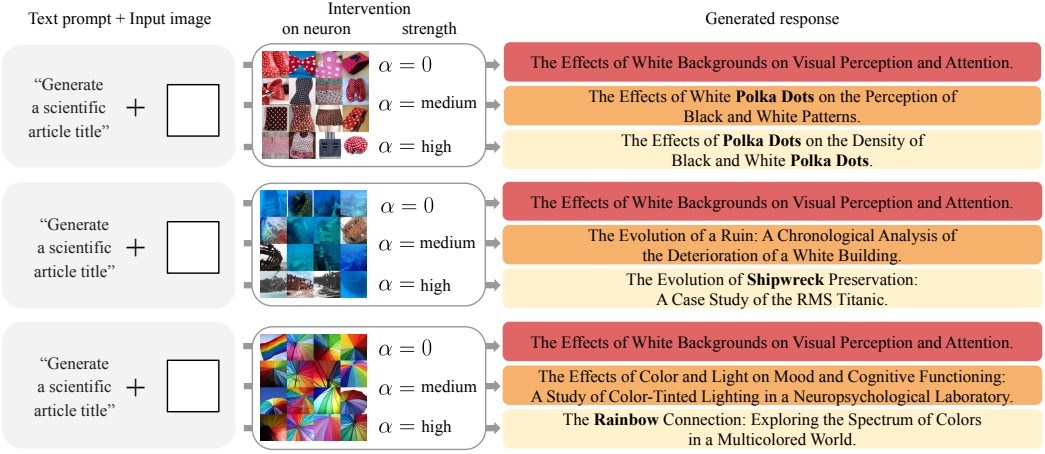

Figure A2: Effects of neuron interventions on MLLM-generated scientific article titles. Steering magnitudes are categorized as "0", "medium", and "high" based on the intervention strength. The neurons are visualized with the highest activating images from which we deduce their associated concepts: "polka dots", "shipwreck", and "rainbow".

The steering capabilities discussed in Section 4.3 are evaluated using an LLM-as-judge setup with the following prompts:

- "Write me a short love poem,"
- "Generate a scientific article title,"
- "Give me a four-item to-do list,"
- "Write me a two-verse rap song,"
- "Propose a math word problem,"
- "Write a paragraph from a Wikipedia page,"
- "Invent a new holiday,"
- "Write a dialogue,"
- "Write a newspaper headline and first paragraph,"
- "Give a conversation starter for a party."

# C User study

To validate the alignment of our MonoSemanticity score (MS) with human judgment, we conducted a user study. Example questions are shown in Figures A3, A4, and A5. Each question shows two grids of images:

$$\left( (x_i)_{i=1}^{16}, \ (y_i)_{i=1}^{16} \right),$$

where each grid contains the 16 images with the highest activations for two neurons $k_x$ and $k_y$, respectively. Formally, for each $i = 1, \ldots, 16$,

$$x_i := \mathbf{x}_n \in \mathcal{I}, \quad \text{rank}_n(a_n^{k_x}) = i,$$

$$y_i := \mathbf{x}_m \in \mathcal{I}, \quad \text{rank}_m(a_m^{k_y}) = i,$$

where $a_n^k$ is the activation of neuron $k$ on image $\mathbf{x}_n$, and $\text{rank}_n(a_n^k)$ is the rank of image $\mathbf{x}_n$ when sorting all images by their activation values in descending order. Images $\mathcal{I}$ come from training set of the ImageNet.

For each neuron pair $(k_x, k_y)$, we asked three human annotators the question: *"Which set of images looks more similar and focused on the same thing?"* Each annotator gave an answer $r_j \in \{k_x, k_y\}$ for $j = 1, 2, 3$. The final human choice was decided by majority vote:

$$R^{\text{user}}_{(k_x, k_y)} \in \{k_x, k_y\}.$$

At the same time, we answered the question using Monosemanticity Score:

$$R^{\text{MS}}_{(k_x, k_y)} := \begin{cases} k_x & \text{if } \text{MS}^{k_x} > \text{MS}^{k_y} \\ k_y & \text{otherwise} \end{cases}$$

We say the MS and users are *aligned* if their decision is the same:

$$\delta_{(k_x, k_y)} := \begin{cases} 1 & \text{if } R^{\text{user}}_{(k_x, k_y)} = R^{\text{MS}}_{(k_x, k_y)} \\ 0 & \text{otherwise} \end{cases}$$

The overall *alignment score* is the fraction of all neuron pairs where the MS and humans are aligned:

$$\text{Alignment Score} = \frac{1}{|\mathcal{Q}|} \sum_{(k_x, k_y) \in \mathcal{Q}} \delta_{(k_x, k_y)},$$

where $\mathcal{Q}$ is the set of all neuron pairs evaluated.

In total, we collected 1,000 user pair rankings with the help of 71 annotators on the Mechanical Turk platform. The number of answers per annotator ranged from 1 to 205, with a median of 24. Annotators were compensated at a rate of \$0.02 per answer.

The neurons used in the study were randomly selected from the last layer of CLIP ViT-L, BatchTopK SAE ($\varepsilon = 4, K = 20$) trained on the last layer of CLIP ViT-L, Matryoshka SAE ($\varepsilon = 4, K = 20$) trained on the last layer of CLIP ViT-L, and BatchTopK SAE ($\varepsilon = 4, K = 20$) trained on the last layer of SigLIP SoViT-400m.

In addition to the plot presenting the user study results in the main paper, we also provide Table A1, which reports the exact values obtained, along with the sizes of each group categorized by MS distances between neuron pairs. When designing the questions, we balanced the number of pairs within each distance interval. Our goal is to evaluate MS computed using embeddings from two different image encoders $E$, namely DINOv2 ViT-B and CLIP ViT-B. As a result, the group sizes are not perfectly equal due to necessary trade-offs. Nevertheless, all groups are sufficiently large and of comparable size.

Table A1: Alignment Scores (AS) obtained from user study. To compute the MS, we use embeddings of image encoder $E$, either DINOv2 ViT-B or CLIP ViT-B. Results are grouped by MS distance between neurons in the question. We made sure that every group is represented by enough pairs.

(a) MS distances computed using DinoV2 embeddings.

| MS Distance (based on DinoV2) | 0.0-0.1 | 0.1-0.2 | 0.2-0.3 | 0.3-0.4 | 0.4-0.5 | 0.5-0.6 | 0.6-0.7 | 0.7-0.8 | 0.8-0.9 |
|---|---|---|---|---|---|---|---|---|---|
| Number of pairs | 177 | 139 | 126 | 138 | 121 | 90 | 105 | 63 | 41 |
| AS ($E$ = DINOv2 ViT-B) | 0.56 | 0.66 | 0.71 | 0.81 | 0.85 | 0.93 | 0.94 | 0.96 | 1.00 |
| AS ($E$ = CLIP ViT-B) | 0.60 | 0.66 | 0.74 | 0.82 | 0.87 | 0.93 | 0.94 | 0.96 | 1.00 |

(b) MS distances computed using CLIP embeddings.

| MS Distance (based on CLIP) | 0.0-0.1 | 0.1-0.2 | 0.2-0.3 | 0.3-0.4 | 0.4-0.5 |
|---|---|---|---|---|---|
| Number of pairs | 292 | 249 | 186 | 178 | 95 |
| AS ($E$ = CLIP ViT-B) | 0.55 | 0.81 | 0.93 | 0.96 | 0.93 |
| AS ($E$ = DINOv2 ViT-B) | 0.53 | 0.77 | 0.92 | 0.96 | 0.93 |

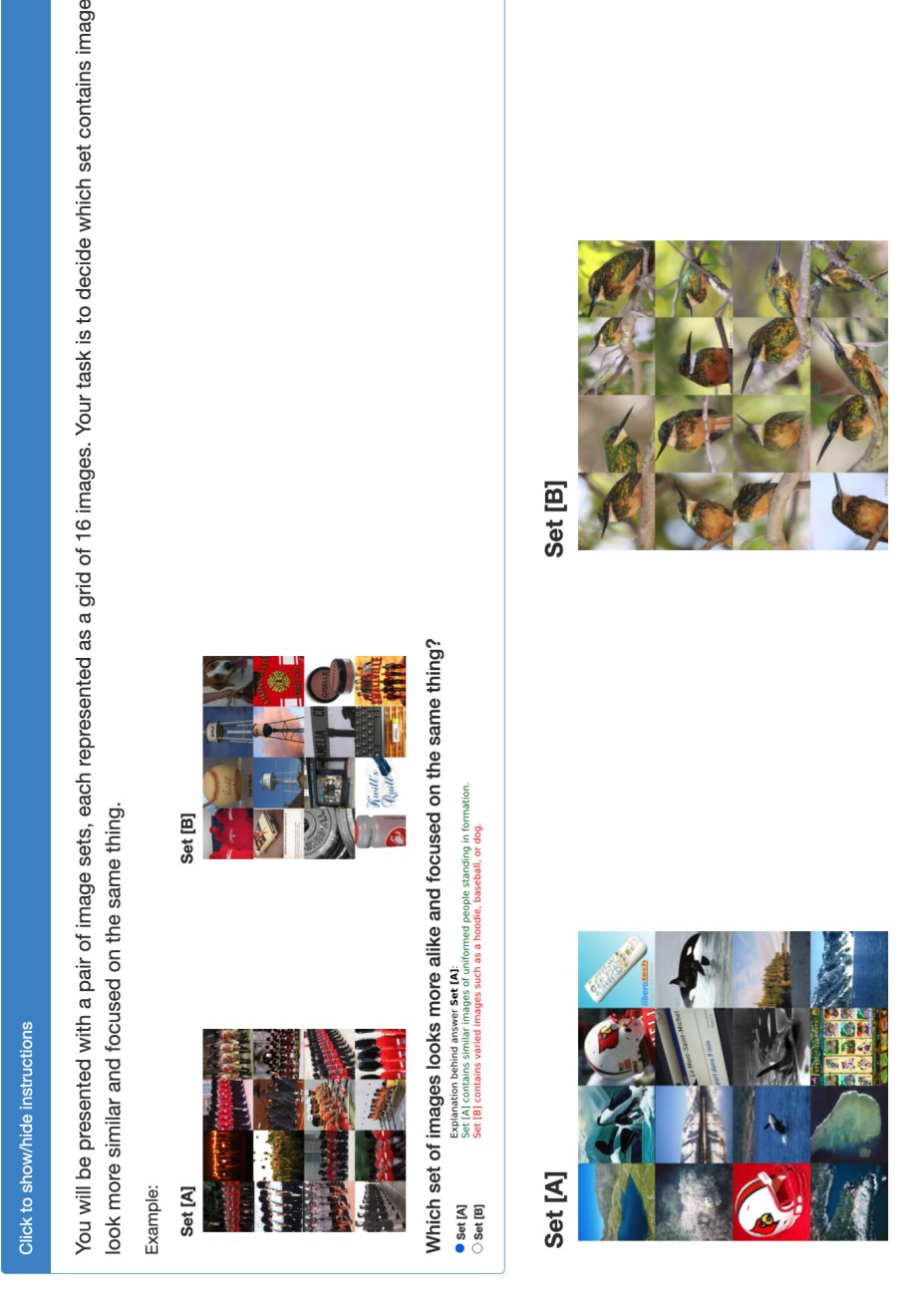

Figure A3: Example question used in the user study. Best viewed horizontally.

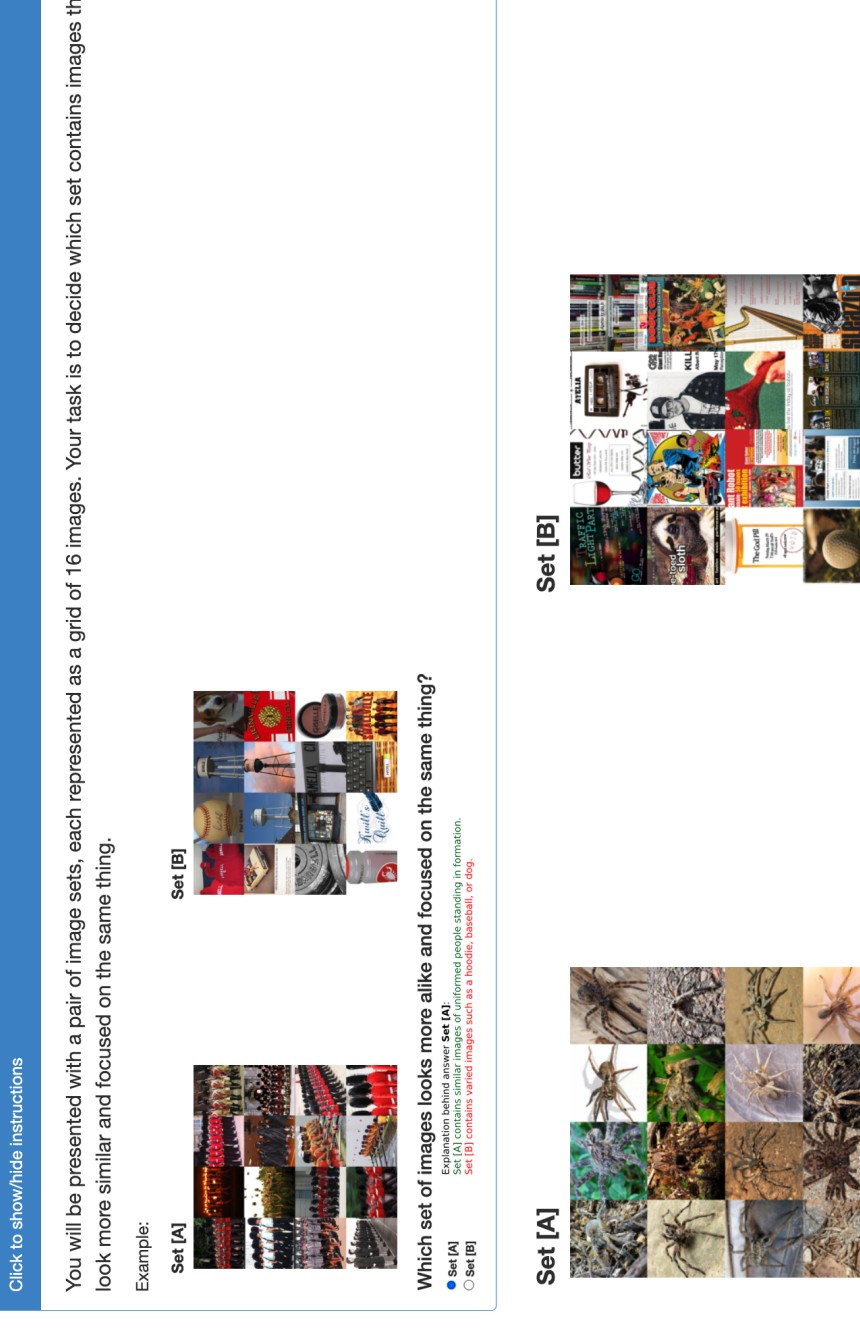

Figure A4: Example question used in the user study. Best viewed horizontally.

You will be presented with a pair of image sets, each represented as a grid of 16 images. Your task is to decide which set contains images that look more similar and focused on the same thing.

Example:

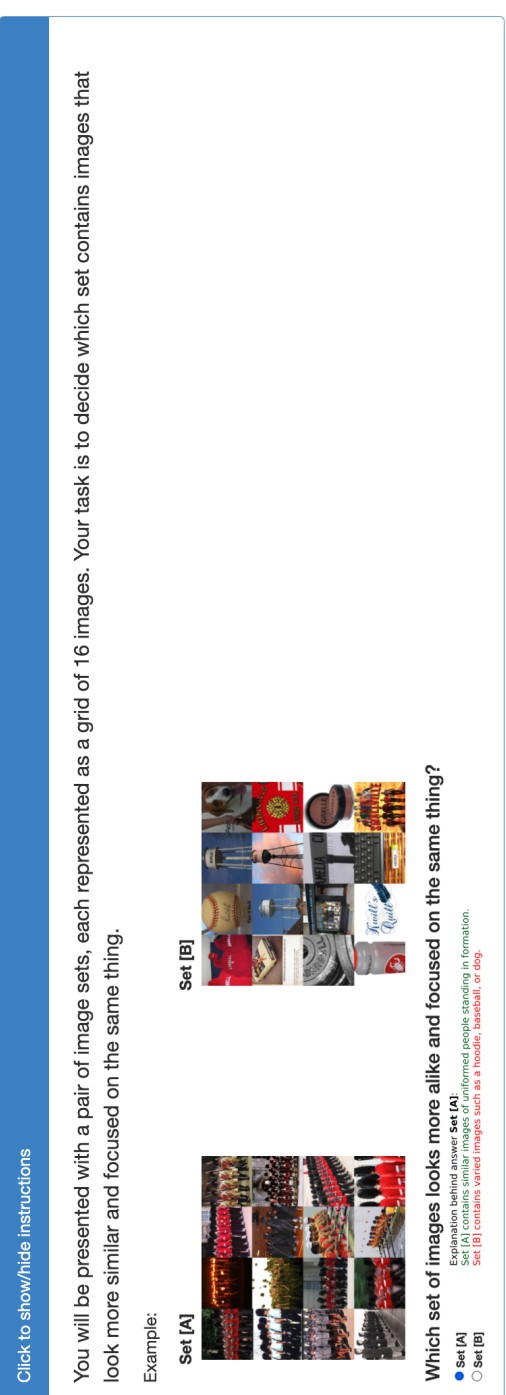

**Which set of images looks more alike and focused on the same thing?**

● Set [A]
○ Set [B]

Explanation behind answer **Set [A]:**
Set [A] contains similar images of uniformed people standing in formation.
Set [B] contains varied images such as a hoodie, baseball, or dog.

Set [A]

Set [B]

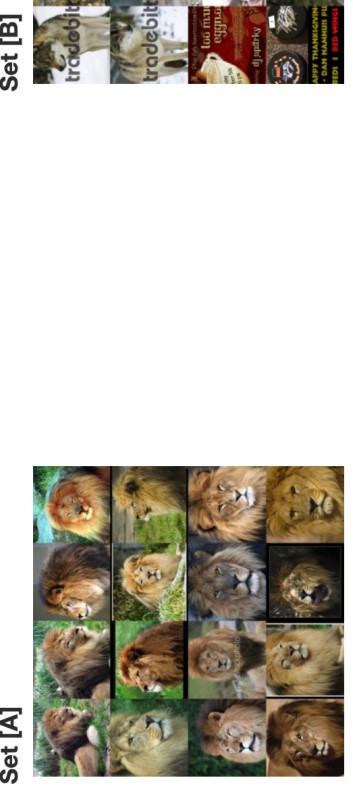

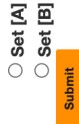

**Which set of images looks more alike and focused on the same thing?**

○ Set [A]
○ Set [B]

Submit

Figure A5: Example question used in the user study. Best viewed horizontally.

## D Benchmark

While MS shows very good results in our user study, we anticipate the development of improved alternatives in the future. To facilitate such advancements, we will release our collected data as a benchmark for evaluating neuron monosemanticity.

The benchmark will include the following files:

- `pairs.csv` – Contains 1000 pairs of neurons $(r_x, r_y)$, along with user preferences $R^{\text{user}}_{(k_x, k_y)}$ and MS values computed using two different image encoders: DINOv2 ViT-B and CLIP ViT-B. Each row includes the following columns: `k_x, k_y, R_user, MS_x_dino, MS_y_dino, MS_x_clip, MS_y_clip`.
- `top16_images.csv` – Lists the 16 most activating images from the ImageNet training set for each neuron used in the study. Columns: `k, x_1, ..., x_16`.
- `activations.csv` – Provides activation values of all 50,000 ImageNet validation images for each neuron. Columns: `k, a_1, ..., a_50000`.

With this data and by following our evaluation procedure, researchers will be able to compare their methods directly to MS under same conditions. They will have access to the same underlying information, specifically the complete set of neuron activations on the ImageNet validation set.

## E Additional results on monosemanticity

### E.1 Unnormalized plots

Monosemanticity scores across all neurons, without normalized index, are shown in Figure A6. We observe that neurons cover a wider range of scores as we increase the width of the SAE layer. Furthermore, for a given threshold of monosemanticity, the number of neurons having a score higher than this threshold is also increasing with the width.

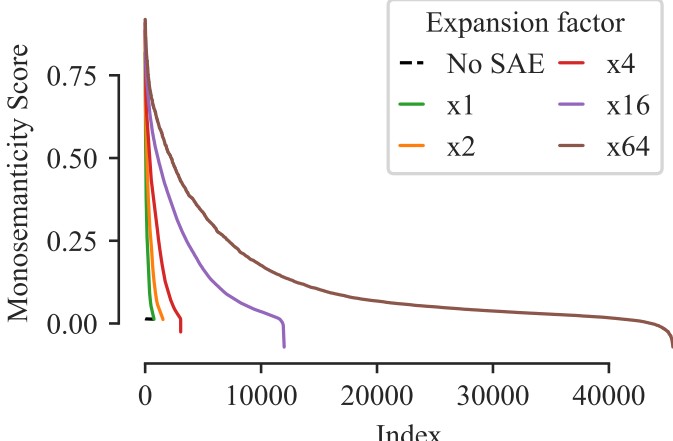

Figure A6: MS in decreasing order across neurons. Results are shown for a layer without SAE ("No SAE"), and with SAE using different expansion factors ($\times 1$, $\times 2$, $\times 4$, $\times 16$ and $\times 64$).

## E.2 Detailed statistics and more models

We report in Tables A2 and A5, the average ($\pm$ std), best and worst monosemanticity scores across neurons for the two SAE variants, attached at different layers and for increasing expansion factors. Although average scores remain similar when increasing expansion factor, we observe a high increase between the original layer and an SAE with expansion factor $\varepsilon = 1$. The best scores get consistentely better as expansion factor gets increased.

Until now, our analysis has focused on SAEs trained on CLIP ViT-L activations, evaluated using the MS score computed from embeddings produced by the DINOv2 image encoder $E$. To broaden this investigation, we now consider SAEs trained on activations from SigLIP SoViT-400m. As an alternative image encoder $E$, we adopt CLIP ViT-B for evaluation.

Tables A3 and A6 show average, best and worst MS computed using CLIP ViT-B as the vision encoder $E$. Even though less distinctively than in original setup, the neurons from SAEs still score better compared to the ones originally found in the model.

In Tables A4 and A7, we report MS statistics for SAEs trained for SigLIP SoViT-400m model computed using CLIP ViT-B as the vision encoder $E$. The results highly resemble the ones for CLIP ViT-L model.

Table A2: The average MS of neurons in a CLIP ViT-L model. DINOv2 ViT-B is used as the image encoder $E$.

| SAE type | Layer | No SAE | Expansion factor | | | | | |
| --- | --- | --- | --- | --- | --- | --- | --- | --- |
| | | | x1 | x2 | x4 | x8 | x16 | x64 |
| BatchTopK | 11 | $0.0135 \pm 0.0003$ | $0.03 \pm 0.06$ | $0.04 \pm 0.06$ | $0.04 \pm 0.06$ | $0.03 \pm 0.05$ | $0.03 \pm 0.05$ | $0.03 \pm 0.05$ |
| | 17 | $0.0135 \pm 0.0004$ | $0.05 \pm 0.07$ | $0.07 \pm 0.09$ | $0.08 \pm 0.11$ | $0.07 \pm 0.10$ | $0.07 \pm 0.10$ | $0.06 \pm 0.10$ |
| | 22 | $0.0135 \pm 0.0003$ | $0.14 \pm 0.12$ | $0.18 \pm 0.15$ | $0.20 \pm 0.17$ | $0.21 \pm 0.17$ | $0.21 \pm 0.18$ | $0.17 \pm 0.18$ |
| | 23 | $0.0135 \pm 0.0003$ | $0.15 \pm 0.13$ | $0.18 \pm 0.16$ | $0.20 \pm 0.17$ | $0.21 \pm 0.17$ | $0.20 \pm 0.18$ | $0.17 \pm 0.18$ |
| | last | $0.0135 \pm 0.0002$ | $0.12 \pm 0.11$ | $0.17 \pm 0.15$ | $0.19 \pm 0.17$ | $0.19 \pm 0.16$ | $0.16 \pm 0.16$ | $0.13 \pm 0.15$ |
| Matryoshka | 11 | $0.0135 \pm 0.0003$ | $0.05 \pm 0.10$ | $0.06 \pm 0.10$ | $0.05 \pm 0.09$ | $0.05 \pm 0.09$ | $0.04 \pm 0.08$ | $0.03 \pm 0.06$ |
| | 17 | $0.0135 \pm 0.0004$ | $0.09 \pm 0.14$ | $0.10 \pm 0.15$ | $0.11 \pm 0.16$ | $0.11 \pm 0.15$ | $0.10 \pm 0.15$ | $0.06 \pm 0.10$ |
| | 22 | $0.0135 \pm 0.0003$ | $0.17 \pm 0.17$ | $0.21 \pm 0.18$ | $0.23 \pm 0.19$ | $0.23 \pm 0.19$ | $0.23 \pm 0.19$ | $0.18 \pm 0.19$ |
| | 23 | $0.0135 \pm 0.0003$ | $0.17 \pm 0.16$ | $0.21 \pm 0.19$ | $0.22 \pm 0.18$ | $0.22 \pm 0.18$ | $0.20 \pm 0.18$ | $0.12 \pm 0.16$ |
| | last | $0.0135 \pm 0.0002$ | $0.16 \pm 0.17$ | $0.20 \pm 0.18$ | $0.23 \pm 0.19$ | $0.22 \pm 0.19$ | $0.19 \pm 0.19$ | $0.13 \pm 0.16$ |

Table A3: The average MS of neurons in a CLIP ViT-L model. CLIP ViT-B is used as the image encoder $E$.

| SAE type | Layer | No SAE | Expansion factor | | | | | |
| --- | --- | --- | --- | --- | --- | --- | --- | --- |
| | | | x1 | x2 | x4 | x8 | x16 | x64 |
| BatchTopK | 11 | $0.4837 \pm 0.0067$ | $0.52 \pm 0.05$ | $0.53 \pm 0.06$ | $0.53 \pm 0.05$ | $0.53 \pm 0.05$ | $0.53 \pm 0.05$ | $0.53 \pm 0.06$ |
| | 17 | $0.4840 \pm 0.0079$ | $0.55 \pm 0.07$ | $0.56 \pm 0.08$ | $0.57 \pm 0.08$ | $0.56 \pm 0.05$ | $0.56 \pm 0.08$ | $0.56 \pm 0.09$ |
| | 22 | $0.4816 \pm 0.0053$ | $0.60 \pm 0.09$ | $0.61 \pm 0.09$ | $0.62 \pm 0.09$ | $0.63 \pm 0.09$ | $0.62 \pm 0.10$ | $0.60 \pm 0.11$ |
| | 23 | $0.4814 \pm 0.0045$ | $0.60 \pm 0.09$ | $0.61 \pm 0.10$ | $0.62 \pm 0.10$ | $0.62 \pm 0.10$ | $0.61 \pm 0.10$ | $0.59 \pm 0.12$ |
| | last | $0.4812 \pm 0.0042$ | $0.59 \pm 0.08$ | $0.60 \pm 0.10$ | $0.61 \pm 0.10$ | $0.61 \pm 0.10$ | $0.59 \pm 0.10$ | $0.56 \pm 0.10$ |
| Matryoshka | 11 | $0.4837 \pm 0.0067$ | $0.54 \pm 0.08$ | $0.55 \pm 0.08$ | $0.55 \pm 0.08$ | $0.54 \pm 0.08$ | $0.53 \pm 0.07$ | $0.52 \pm 0.06$ |
| | 17 | $0.4840 \pm 0.0079$ | $0.57 \pm 0.09$ | $0.58 \pm 0.09$ | $0.58 \pm 0.10$ | $0.58 \pm 0.10$ | $0.57 \pm 0.10$ | $0.54 \pm 0.09$ |
| | 22 | $0.4816 \pm 0.0053$ | $0.61 \pm 0.09$ | $0.62 \pm 0.09$ | $0.63 \pm 0.10$ | $0.62 \pm 0.11$ | $0.62 \pm 0.11$ | $0.59 \pm 0.12$ |
| | 23 | $0.4814 \pm 0.0045$ | $0.60 \pm 0.09$ | $0.62 \pm 0.10$ | $0.62 \pm 0.10$ | $0.61 \pm 0.11$ | $0.60 \pm 0.11$ | $0.54 \pm 0.11$ |
| | last | $0.4812 \pm 0.0042$ | $0.59 \pm 0.09$ | $0.61 \pm 0.10$ | $0.62 \pm 0.11$ | $0.61 \pm 0.11$ | $0.59 \pm 0.12$ | $0.54 \pm 0.12$ |

Table A4: The average MS of neurons in a SigLIP SoViT-400m model. CLIP ViT-B is used as the image encoder $E$.

| SAE type | Layer | No SAE | Expansion factor | | | | | |
|---|---|---|---|---|---|---|---|---|
| | | | x1 | x2 | x4 | x8 | x16 | x64 |
| BatchTopK | 11 | 0.4805 ± 0.0014 | 0.50 ± 0.03 | 0.51 ± 0.04 | 0.51 ± 0.05 | 0.51 ± 0.06 | 0.52 ± 0.06 | 0.52 ± 0.07 |
| | 16 | 0.4809 ± 0.0024 | 0.51 ± 0.04 | 0.52 ± 0.05 | 0.52 ± 0.06 | 0.53 ± 0.07 | 0.53 ± 0.07 | 0.53 ± 0.08 |
| | 21 | 0.4810 ± 0.0052 | 0.52 ± 0.05 | 0.53 ± 0.06 | 0.53 ± 0.06 | 0.53 ± 0.07 | 0.54 ± 0.08 | 0.53 ± 0.08 |
| | last | 0.4811 ± 0.0048 | 0.61 ± 0.09 | 0.61 ± 0.09 | 0.62 ± 0.09 | 0.62 ± 0.09 | 0.62 ± 0.10 | 0.60 ± 0.11 |
| Matryoshka | 11 | 0.4805 ± 0.0014 | 0.50 ± 0.03 | 0.50 ± 0.05 | 0.50 ± 0.05 | 0.50 ± 0.06 | 0.51 ± 0.07 | 0.51 ± 0.07 |
| | 16 | 0.4809 ± 0.0024 | 0.51 ± 0.05 | 0.52 ± 0.06 | 0.52 ± 0.07 | 0.52 ± 0.07 | 0.52 ± 0.07 | 0.51 ± 0.07 |
| | 21 | 0.4810 ± 0.0052 | 0.52 ± 0.05 | 0.53 ± 0.06 | 0.53 ± 0.06 | 0.53 ± 0.07 | 0.52 ± 0.07 | 0.51 ± 0.07 |
| | last | 0.4811 ± 0.0048 | 0.61 ± 0.09 | 0.62 ± 0.10 | 0.62 ± 0.10 | 0.62 ± 0.10 | 0.60 ± 0.11 | 0.58 ± 0.11 |

Table A5: Comparison of the best / worst MS of neurons in a CLIP ViT-L model. DINOv2 ViT-B is used as the image encoder $E$.

| SAE type | Layer | No SAE | Expansion factor | | | | | |
|---|---|---|---|---|---|---|---|---|
| | | | ×1 | ×2 | ×4 | ×8 | ×16 | ×64 |
| BatchTopK | 11 | 0.01 / 0.01 | 0.61 / -0.02 | 0.73 / -0.08 | 0.71 / -0.06 | 0.87 / -0.07 | 0.90 / -0.10 | 1.00 / -0.11 |
| | 17 | 0.01 / 0.01 | 0.65 / 0.01 | 0.79 / -0.02 | 0.86 / -0.07 | 0.86 / -0.08 | 0.93 / -0.08 | 1.00 / -0.12 |
| | 22 | 0.01 / 0.01 | 0.66 / 0.01 | 0.79 / 0.01 | 0.80 / 0.01 | 0.88 / -0.08 | 0.92 / -0.06 | 1.00 / -0.11 |
| | 23 | 0.01 / 0.01 | 0.73 / 0.01 | 0.72 / 0.01 | 0.83 / 0.01 | 0.89 / -0.02 | 0.93 / -0.06 | 1.00 / -0.10 |
| | last | 0.01 / 0.01 | 0.57 / 0.01 | 0.78 / 0.01 | 0.78 / 0.01 | 0.81 / -0.01 | 0.85 / -0.04 | 1.00 / -0.10 |
| Matryoshka | 11 | 0.01 / 0.01 | 0.84 / -0.06 | 0.90 / -0.07 | 0.95 / -0.08 | 1.00 / -0.11 | 0.89 / -0.10 | 1.00 / -0.10 |
| | 17 | 0.01 / 0.01 | 0.86 / -0.04 | 0.84 / -0.05 | 0.93 / -0.07 | 0.94 / -0.09 | 0.96 / -0.08 | 1.00 / -0.14 |
| | 22 | 0.01 / 0.01 | 0.83 / 0.01 | 0.83 / 0.01 | 0.87 / -0.02 | 0.94 / -0.06 | 1.00 / -0.11 | 1.00 / -0.11 |
| | 23 | 0.01 / 0.01 | 0.82 / 0.01 | 0.84 / 0.01 | 0.89 / -0.04 | 0.93 / -0.04 | 0.96 / -0.06 | 1.00 / -0.11 |
| | last | 0.01 / 0.01 | 0.82 / 0.01 | 0.91 / 0.01 | 0.89 / -0.03 | 0.93 / -0.05 | 0.91 / -0.07 | 1.00 / -0.12 |

Table A6: Comparison of the best / worst MS of neurons in a CLIP ViT-Large model. CLIP ViT-B is used as the image encoder $E$.

| SAE type | Layer | No SAE | Expansion factor | | | | | |
|---|---|---|---|---|---|---|---|---|
| | | | ×1 | ×2 | ×4 | ×8 | ×16 | ×64 |
| BatchTopK | 11 | 0.50 / 0.47 | 0.80 / 0.41 | 0.87 / 0.38 | 0.90 / 0.28 | 0.91 / 0.27 | 0.95 / 0.24 | 1.00 / 0.20 |
| | 17 | 0.50 / 0.47 | 0.84 / 0.37 | 0.87 / 0.33 | 0.94 / 0.35 | 0.94 / 0.28 | 0.96 / 0.24 | 1.00 / 0.14 |
| | 22 | 0.50 / 0.47 | 0.82 / 0.39 | 0.85 / 0.38 | 0.89 / 0.37 | 0.93 / 0.29 | 0.93 / 0.15 | 1.00 / 0.15 |
| | 23 | 0.50 / 0.47 | 0.81 / 0.41 | 0.84 / 0.40 | 0.89 / 0.35 | 0.91 / 0.27 | 0.93 / 0.24 | 1.00 / 0.08 |
| | last | 0.50 / 0.47 | 0.80 / 0.40 | 0.84 / 0.40 | 0.87 / 0.36 | 0.87 / 0.31 | 0.89 / 0.25 | 1.00 / 0.17 |
| Matryoshka | 11 | 0.50 / 0.47 | 0.90 / 0.39 | 0.95 / 0.31 | 0.97 / 0.23 | 1.00 / 0.22 | 0.94 / 0.18 | 1.00 / 0.19 |
| | 17 | 0.50 / 0.47 | 0.94 / 0.33 | 0.93 / 0.35 | 0.96 / 0.29 | 0.96 / 0.22 | 0.97 / 0.14 | 1.00 / 0.11 |
| | 22 | 0.50 / 0.47 | 0.88 / 0.40 | 0.87 / 0.33 | 0.89 / 0.29 | 0.94 / 0.23 | 1.00 / 0.15 | 1.00 / 0.06 |
| | 23 | 0.50 / 0.47 | 0.85 / 0.40 | 0.86 / 0.35 | 0.90 / 0.35 | 0.91 / 0.19 | 0.93 / 0.17 | 1.00 / 0.14 |
| | last | 0.50 / 0.47 | 0.85 / 0.41 | 0.88 / 0.40 | 0.89 / 0.31 | 0.91 / 0.26 | 0.92 / 0.17 | 1.00 / 0.09 |

Table A7: Comparison of the best / worst MS of neurons in a SigLIP SoViT-400m model. CLIP ViT-B is used as the image encoder $E$.

| SAE type | Layer | No SAE | Expansion factor | | | | | |
|---|---|---|---|---|---|---|---|---|
| | | | ×1 | ×2 | ×4 | ×8 | ×16 | ×64 |
| BatchTopK | 11 | 0.49 / 0.48 | 0.61 / 0.41 | 0.83 / 0.29 | 0.88 / 0.27 | 0.90 / 0.23 | 1.00 / 0.12 | 1.00 / 0.15 |
| | 16 | 0.53 / 0.47 | 0.74 / 0.38 | 0.75 / 0.34 | 0.93 / 0.25 | 0.94 / 0.20 | 0.93 / 0.22 | 1.00 / 0.18 |
| | 21 | 0.54 / 0.47 | 0.76 / 0.38 | 0.77 / 0.35 | 0.83 / 0.25 | 0.89 / 0.17 | 0.95 / 0.20 | 1.00 / 0.11 |
| | last | 0.50 / 0.47 | 0.83 / 0.41 | 0.86 / 0.40 | 0.88 / 0.37 | 0.92 / 0.33 | 0.93 / 0.20 | 1.00 / 0.11 |
| Matryoshka | 11 | 0.49 / 0.48 | 0.70 / 0.40 | 0.93 / 0.29 | 0.77 / 0.27 | 0.93 / 0.18 | 0.91 / 0.22 | 1.00 / 0.16 |
| | 16 | 0.53 / 0.47 | 0.78 / 0.40 | 0.84 / 0.29 | 0.91 / 0.19 | 0.93 / 0.18 | 1.00 / 0.19 | 1.00 / 0.16 |
| | 21 | 0.54 / 0.47 | 0.85 / 0.39 | 0.81 / 0.37 | 0.83 / 0.25 | 0.93 / 0.24 | 0.94 / 0.21 | 1.00 / 0.15 |
| | last | 0.50 / 0.47 | 0.87 / 0.40 | 0.87 / 0.38 | 0.89 / 0.30 | 0.91 / 0.25 | 0.94 / 0.15 | 1.00 / 0.15 |

In Figure A7 we plot MS across single neurons. We consider setups in which (a) neurons of CLIP ViT-L are evaluated with DINOv2 as the image encoder $E$, (b) neurons of CLIP ViT-L are evaluated with CLIP ViT-B as $E$, and (c) neurons of SigLIP SoViT-400m are evaluated with CLIP ViT-B as $E$. In all three cases SAE neurons are more monosemantic compared to the original neurons of the models. It shows that MS results are consistent across different architectures being both explained and used as $E$.

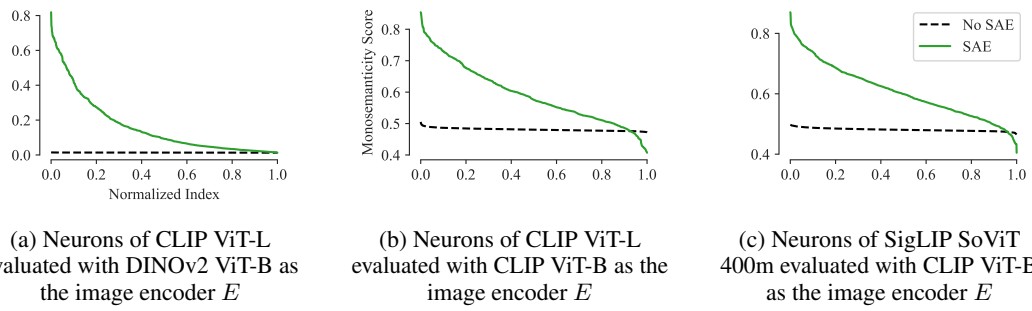

(a) Neurons of CLIP ViT-L evaluated with DINOv2 ViT-B as the image encoder $E$

(b) Neurons of CLIP ViT-L evaluated with CLIP ViT-B as the image encoder $E$

(c) Neurons of SigLIP SoViT 400m evaluated with CLIP ViT-B as the image encoder $E$

Figure A7: MS in decreasing order across neurons. Results are shown for the last layers of two different models, without SAE (black dashed line), and with SAE being trained with expansion factor 1 (green solid line). MS is computed with distinct image encoders $E$.

In Figures A8 and A9, we plot again MS scores across neurons for SAEs trained with different expansion factors and sparsity levels, but using CLIP ViT-B as the image encoder $E$. We observe very similar patterns when compared to the MS computed using DINOv2 ViT-B. Both higher expansion factor and lower sparsity helps find more of the monosemantic units.

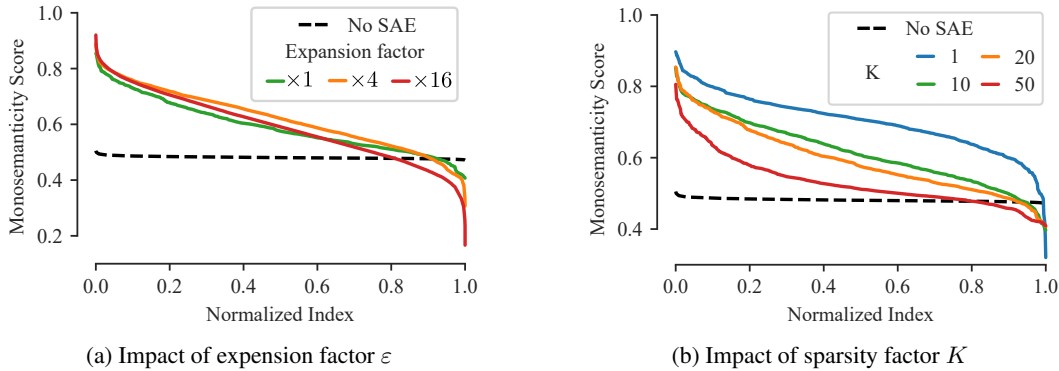

(a) Impact of expension factor $\varepsilon$

(b) Impact of sparsity factor $K$

Figure A8: Monosemanticity Scores (computed using CLIP ViT-B) in decreasing order across neurons, normalized by width. Results are shown for the last layer of the model, without SAE ("No SAE", in black dashed line), and with SAE using either (a) different expansion factors (in straight lines, for $\varepsilon = 1$, for $\varepsilon = 4$ and for $\varepsilon = 16$) or (b) different sparsity levels, with straight lines for $K = 1$, for $K = 10$, for $K = 20$, and for $K = 50$.

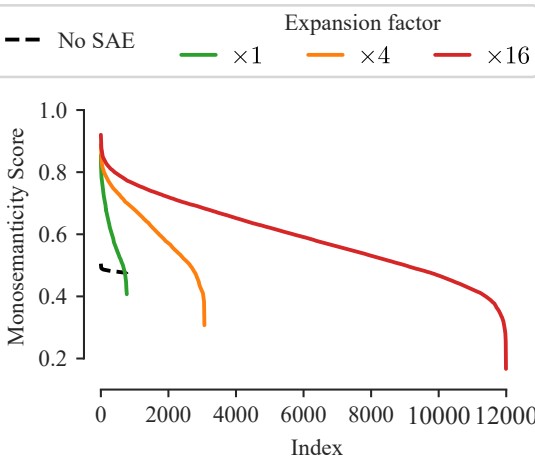

Figure A9: Monosemanticity Scores (computed using CLIP ViT-B) in decreasing order across neurons without normalizing by width. Results are shown for a layer without SAE ("No SAE"), and with SAE using different expansion factors ($\times 1$, $\times 4$ and $\times 16$).

### E.3 Matryoshka hierarchies

We train and evaluate the SAE on embeddings extracted from iNaturalist [55] dataset using an expansion factor $\varepsilon = 2$ and groups of size $\mathcal{M} = \{3, 16, 69, 359, 1536\}$. These group sizes correspond to the numbers of nodes of the first 5 levels of the species taxonomy tree of the dataset, i.e. the respective number of *"kingdoms"*, *"phylums"*, *"classes"*, *"orders"*, and *"families"*.

To measure the granularity of the concepts, we map each neuron to the most fitting depth in the iNaturalist taxonomy tree to compare the hierarchy of concepts within the Matryoshka SAE with human-defined ones. To obtain this neuron-to-depth mapping, we select the top-16 activating images per neuron, and compute the average depth of the Lowest Common Ancestors (LCA) in the taxonomy tree for each pair of images. For instance, given a neuron with an average LCA depth of 2, we can assume that images activating this neuron are associated to species from multiple *"classes"* of the same *"phylum"*. We report the average assigned LCA depth of neurons across the Matryoshka group level in Table A8. We notice that average LCA depths are correlated with the level, suggesting that the Matryoshka hierarchy can be aligned with human-defined hierarchy. We additionally aggregate statistics of MS of neurons for each level. Average and maximum MS also correlates with the level, confirming that the most specialized neurons are found in the lowest levels.

Table A8: Average LCA depth and monosemanticity (MS) scores across neurons at each level in the Matryoshka nested dictionary.

|    | **Level** | **0** | **1** | **2** | **3** | **4** |
|----|-----------|-------|-------|-------|-------|-------|
|    | **Depth** | 3.33  | 2.92  | 3.85  | 3.86  | 4.06  |
| **MS** | **Avg.**  | 0.06  | 0.08  | 0.09  | 0.16  | 0.24  |
|    | **Max.**  | 0.11  | 0.30  | 0.29  | 0.69  | 0.76  |
|    | **Min.**  | 0.04  | 0.03  | 0.03  | 0.03  | -0.05 |

# F Reconstruction of SAEs

In Table A9 and Table A10, we report respectively $R^2$ and sparsity ($L_0$), for the two SAE variants we compare in Section 4.2. As BatchTopK activation enforces sparsity on *batch-level*, during test-time it is replaced with ReLU($\mathbf{x} - \gamma$), with $\mathbf{x}$ is the input and $\gamma$ is a vector of thresholds estimated for each neuron, as the average of the minimum positive activation values across a number of batches. For this reason the test-time sparsity may slightly differ from $K$ fixed at the value of 20 in our case.

We report in Table A11 the detailed metrics ($R^2$, $L_0$ and statistics of MS) obtained for SAEs trained with different $K$ values considered in Section 4.2.

Table A9: Comparison of $R^2$ (in %) by different SAEs trained with $K = 20$ for a CLIP ViT-L model.

| SAE type | Layer | No SAE | Expansion factor | | | | | |
|---|---|---|---|---|---|---|---|---|
| | | | x1 | x2 | x4 | x8 | x16 | x64 |
| BatchTopK | 11 | 100 | 74.7 | 75.0 | 75.1 | 75.0 | 74.7 | 73.5 |
| | 17 | 100 | 70.4 | 71.9 | 72.6 | 72.9 | 72.9 | 72.5 |
| | 22 | 100 | 68.7 | 72.6 | 74.9 | 76.0 | 76.8 | 77.4 |
| | 23 | 100 | 67.2 | 71.5 | 74.0 | 75.3 | 76.0 | 76.8 |
| | last | 100 | 70.1 | 74.6 | 77.1 | 78.2 | 78.6 | 79.1 |
| Matryoshka | 11 | 100 | 72.8 | 73.9 | 74.5 | 75.1 | 75.2 | 74.5 |
| | 17 | 100 | 67.3 | 69.5 | 70.7 | 71.8 | 72.6 | 72.7 |
| | 22 | 100 | 65.5 | 69.6 | 71.5 | 74.0 | 75.4 | 76.6 |
| | 23 | 100 | 63.9 | 68.5 | 71.0 | 73.1 | 74.8 | 74.6 |
| | last | 100 | 66.8 | 71.6 | 74.1 | 76.0 | 77.6 | 78.2 |

Table A10: Comparison of true sparsity measured by $L_0$-norm for different SAEs trained with $K = 20$ for a CLIP ViT-L model.

| SAE type | Layer | No SAE | Expansion factor | | | | | |
|---|---|---|---|---|---|---|---|---|
| | | | x1 | x2 | x4 | x8 | x16 | x64 |
| BatchTopK | 11 | 1024 | 19.7 | 19.5 | 19.4 | 19.6 | 20.0 | 22.9 |
| | 17 | 1024 | 19.4 | 19.4 | 19.2 | 19.6 | 19.5 | 22.3 |
| | 22 | 1024 | 19.6 | 19.7 | 19.7 | 19.8 | 20.3 | 23.0 |
| | 23 | 1024 | 19.8 | 19.8 | 19.9 | 20.1 | 20.3 | 22.2 |
| | last | 768 | 19.9 | 19.9 | 19.9 | 20.1 | 20.2 | 22.2 |
| Matryoshka | 11 | 1024 | 19.4 | 19.5 | 19.4 | 19.6 | 19.8 | 21.3 |
| | 17 | 1024 | 19.3 | 19.3 | 19.3 | 19.4 | 19.5 | 20.5 |
| | 22 | 1024 | 19.7 | 19.7 | 19.6 | 19.8 | 19.9 | 22.0 |
| | 23 | 1024 | 19.7 | 19.8 | 19.8 | 19.9 | 20.6 | 25.1 |
| | last | 768 | 20.0 | 19.9 | 19.8 | 19.9 | 20.2 | 22.5 |

Table A11: Statistics for SAEs trained with different sparsity constraint $K$ on activations of the last layer with expansion factor 16. "No SAE" row contains results for raw activations before attaching the SAE.

| $K$ | $L_0$ | $R^2(\%)$ | MS | | |
|---|---|---|---|---|---|
| | | | Min | Max | Mean |
| 1 | 0.9 | 31.3 | -0.03 | 0.90 | $0.37 \pm 0.20$ |
| 10 | 9.9 | 60.6 | 0.01 | 0.79 | $0.19 \pm 0.16$ |
| 20 | 20.0 | 66.8 | 0.01 | 0.82 | $0.16 \pm 0.17$ |
| 50 | 50.1 | 74.9 | 0.01 | 0.69 | $0.07 \pm 0.08$ |
| No SAE | – | – | 0.01 | 0.01 | $0.01 \pm 0.00$ |

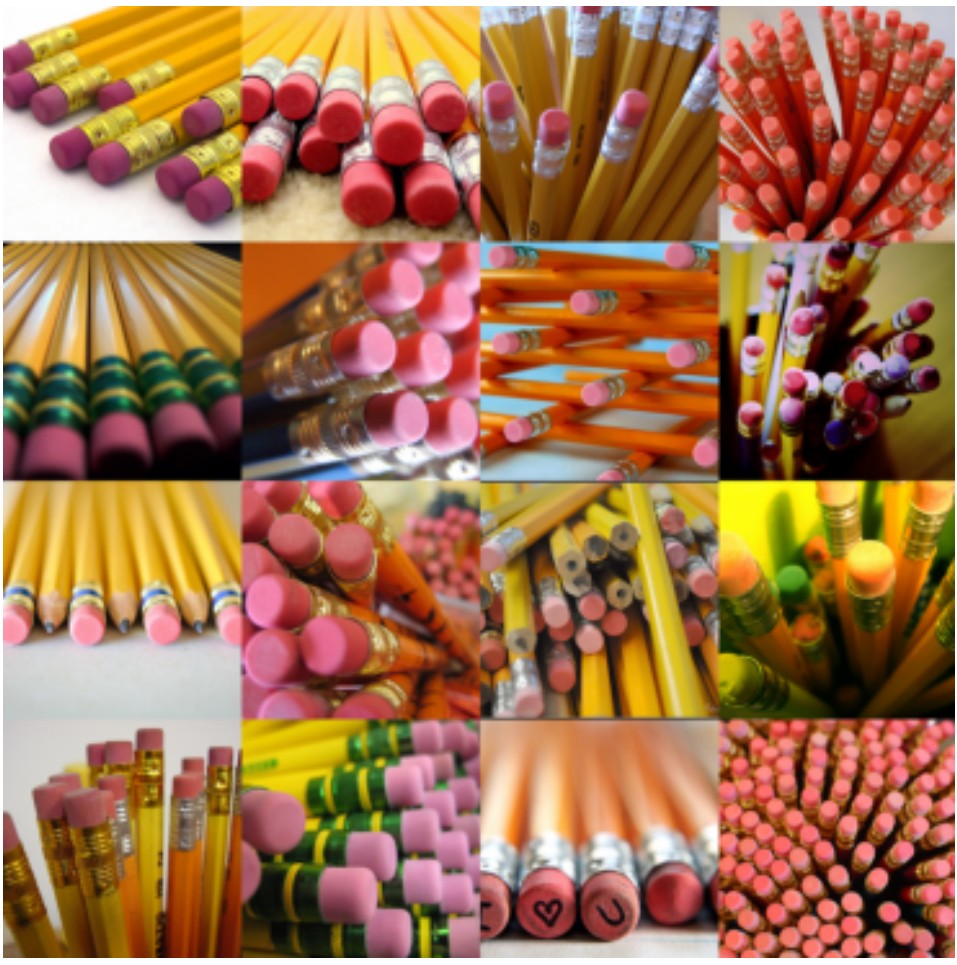

Figure A10: Images highly activating the neuron we intervene on in Figure 6, which we manually labeled as "Pencil Neuron".

## G   Uniqueness of concepts

The sparse reconstruction objective regularizes the SAE activations to focus on different concepts. To confirm it in practice, we collect top-16 highest activating images for each neuron of SAE and compute Jaccard Index $J$ between every pair of neurons. The images come from training set. We exclude 10 out of 12288 neurons for which we found less than 16 activating images and use Matryoshka SAE trained on the last layer with expansion factor of 16. We find that $J > 0$ for 16000 out of 75368503 pairs ($> 0.03\%$) and $J > 0.5$ for only 20 pairs, which shows very high uniqueness of learned concepts.

## H   Additional qualitative results

We illustrate in Figure A10 the highly activating images for the "Pencil" neuron, which we used for steering in Figure 6. In Figures A11 and A12 we provide more randomly selected examples of neurons for which we computed MS using two different image encoders. In both cases we see a clear correlation between score and similarity of images in a grid.

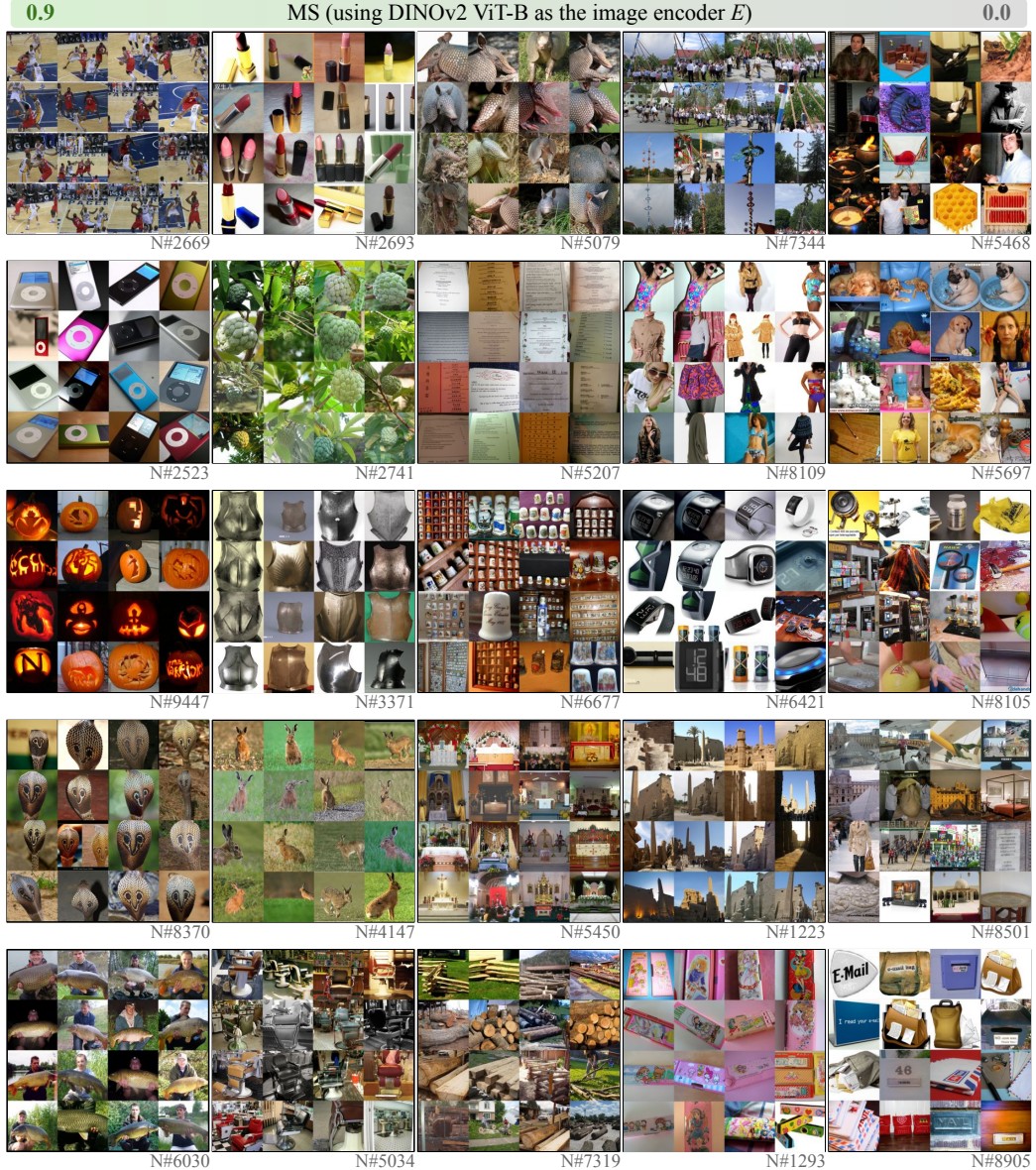

Figure A11: Qualitative examples of highest activating images for different neurons from high (left) to low (right) MS score. As the metric gets higher, highest activating images are more similar, illustrating the correlation with monosemanticity. DINOv2 ViT-B is used as the image encoder $E$.

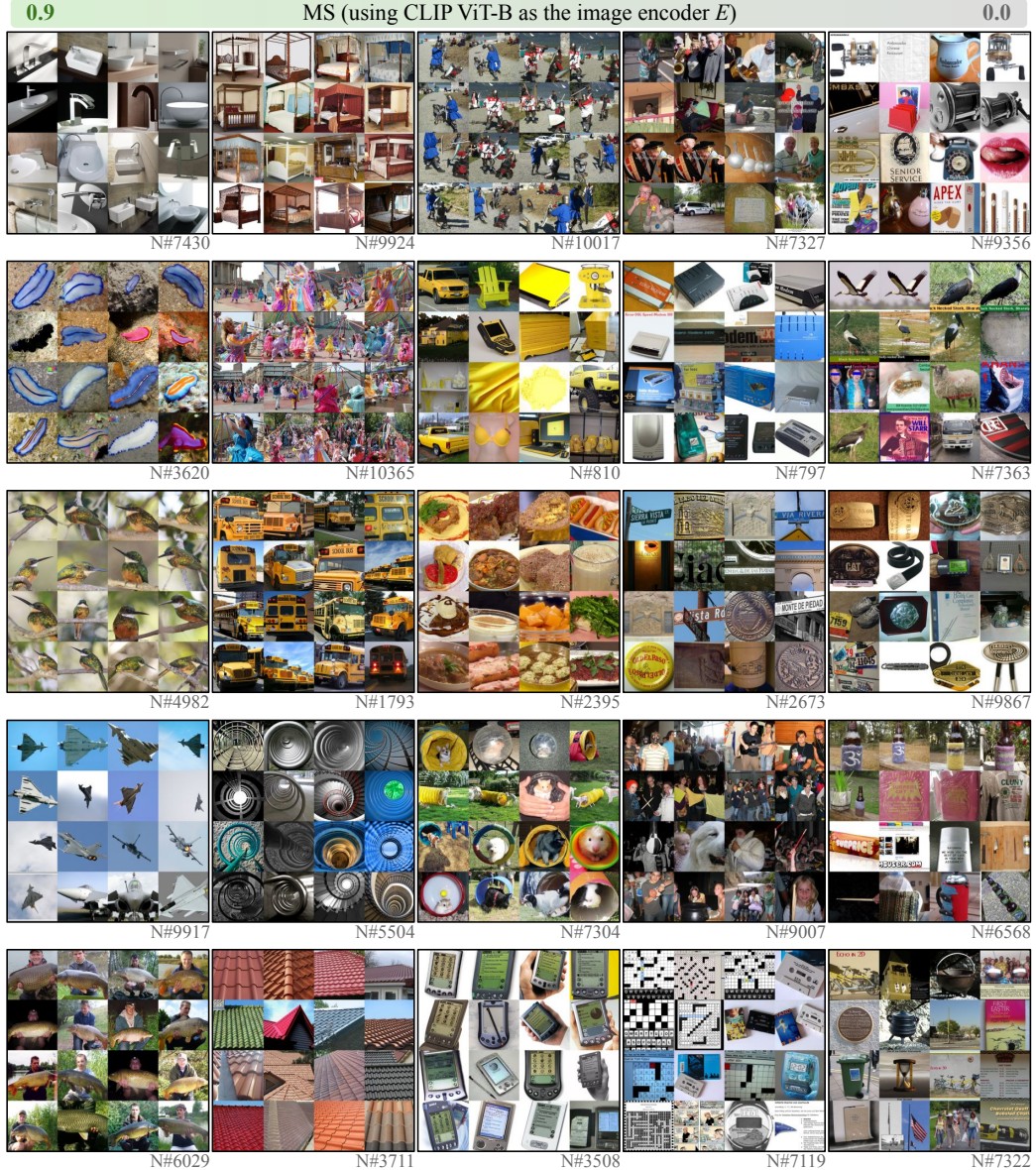

Figure A12: Qualitative examples of highest activating images for different neurons from high (left) to low (right) MS score. As the metric gets higher, highest activating images are more similar, illustrating the correlation with monosemanticity. CLIP ViT-B is used as the image encoder $E$.

