# OpenReview forum: "Sparse Autoencoders Learn Monosemantic Features in Vision-Language Models"
_NeurIPS.cc/2025/Conference — NeurIPS 2025 poster_

### Official Review · Reviewer_6AZV · 2025-06-29

**Clarity:** 3
**Significance:** 2
**Originality:** 2
**Rating:** 4
**Confidence:** 4

**Summary:**

This paper proposes a novel metric for evaluating sparse autoencoders (SAEs) applied to image-text encoders such as CLIP. The metric appears to be correlated with human evaluations, offering a promising direction for interpreting multimodal representations. Additionally, the authors explore how SAE neurons can be used to steer multimodal large language models (VLMs), providing qualitative evidence of controllability. The paper echoes insights from prior work on LLMs, such as the increase in monosemanticity with wider SAEs.

**Questions:**

Please check weakneses.

**Ethical Concerns:**

["NO or VERY MINOR ethics concerns only"]

**Final Justification:**

The authors addressed most of my concerns and hence I am increasing my score.

**Limitations:**

Very short limitation paragraph at the end of the paper. I expect longer discussion on limitations as meantioned in the weakneses section.

**Quality:**

3

**Strengths And Weaknesses:**

**Strengths:**

* The proposal of a metric for evaluating SAEs is needed and important. The metric is intuitive and seems to be correlation with human judgment.

* The qualitative results show that interventions on SAE neurons in vision encoders can meaningfully steer the output of VLMs.

* The paper is well written, clearly structured, and easy to follow.


**Weakneses:**

* Limited scope of encoders: The experiments are limited to CLIP-style encoders (CLIP and SigLIP). It is unclear whether the approach generalizes to other types of vision encoders. Including results on more recent state-of-the-art models (e.g., AIMv2 [1]) would strengthen the contribution.

* The paper does not discuss or compare with other recent approaches to VLM steering, such as activation-based steering methods [2]. Activation steering is a common baseline in LLMs [3] and VLMs. A comparison and discussion with other steering method would properly position this paper.

* The method focuses exclusively on steering the vision encoder. How does this compare with steering interventions applied within the LLM, such as on vision tokens after the connector in multimodal LLMs? A discussion of the trade-offs or benefits would be valuable.

* Lack of steering quantitative evaluation: The paper only presents qualitative results for steering, which makes it difficult to assess practical effectiveness. Quantitative metrics or user studies would provide stronger evidence of the method’s utility.

* The proposed metric includes a relevance score. It would be helpful to show how much this component contributes to overall performance -- e.g., through an ablation comparing versions with and without it.

* The paper could better articulate how the proposed method helps reveal differences between vision encoders. Are there insights into architectural or representational differences?

[1] "Analyzing Fine-tuning Representation Shift for Multimodal LLMs Steering alignment." ICCV (2025).

[2]  "Multimodal autoregressive pre-training of large vision encoders." CVPR (2025).

[3] "Refusal in language models is mediated by a single direction."  NeurIPS (2024).

---

> ### Author Rebuttal · Authors · 2025-07-31
>
> We thank the Reviewer 6AZV for the positive and constructive feedback recognizing our metric as *needed, important, intuitive* and *correlated with human judgment*; and acknowledging our results showing that *interventions on SAE neurons in vision encoders can meaningfully steer the output of VLMs.* We are happy to hear that our paper is *well written, clearly structured, and easy to follow.*
>
> **Generalization to other encoders than just CLIP-style encoders (CLIP and SigLIP).**
>
> We appreciate the reviewer’s suggestion and have conducted additional experiments on state-of-the-art vision encoders, including **AIMv2** [1] and **WebSSL-MAE** [4] and report the best observed MS.
>
> | Vision Encoder   | Without SAE | With SAE (ε=1) | With SAE (ε=4) |
> |---------------|--------|-----------|-----------|
> | AIMv2          		| 0.01   | 0.59      | 0.85      |
> | WebSSL-MAE   	 | 0.01   | 0.79      | 0.92      |
> | CLIP          		| 0.01   | 0.82       | 0.89      |
> | SigLIP        		| 0.01   | 0.83       | 0.88      |
>
> In fact, neurons in these recent encoders are also not monosemantic in their native form, but the SAEs can successfully disentangle them into highly monosemantic units. In addition, we observe that increasing the expansion factor ε of the SAE leads to higher MS scores universally across different encoders. We would be happy to include these results in the final manuscript.
>
> **Discussing other recent approaches to VLM steering, e.g. [2, 3].**
>
> Unlike activation steering methods (e.g., [2], [3]), which typically involve direct manipulation of the language decoder’s residual stream and often require fine-tuning or labeled data, our method operates in a fully unsupervised manner. The SAE is trained independently on the vision encoder using a generic and broad dataset (i.e. ImageNet), and yet it enables effective steering of the overall MLLM. We believe this is a valuable and novel finding, as it demonstrates that meaningful control over multimodal outputs can be achieved without altering the language generation components or relying on annotated vision-language datasets.
> While a direct quantitative comparison with activation-based methods is non-trivial due to the differing mechanisms and representations, we will expand the related work section to include a discussion of these papers.
>
> **How does steering the vision encoder compare with steering interventions applied within the LLM?**
>
> Indeed, we focus on steering the vision encoder, rather than applying interventions within the language model or after the connector. This offers several important advantages: 1) it is more universal, as many MLLMs share common vision encoders (e.g. CLIP, SigLIP), even when their language part differs; 2) it is entirely unsupervised and does not rely on vision-language dataset or labeled data which are typically required for steering within the LLM; 3) it preserves internal language representations and is more interpretable when the goal is to filter visual concepts. We will add this discussion to clarify benefits and limitations of steering at different stages of the MLLM pipeline. We believe our method complements existing approaches and offers an interpretable and broadly applicable alternative for multimodal control.
>
> **Lack of steering quantitative evaluation**
>
> While Table 2 in the main paper demonstrates a basic quantitative signal of effective steering, we agree that a more detailed and interpretable metric is valuable. To this end, we adopt an *LLM-as-a-judge* approach [5], using GPT-4.1-mini to evaluate both concept insertion and suppression.
>
> **Concept insertion.**
> We consider the first 100 SAE neurons $x_i$ used in LLaVA experiments. For each neuron $x_i$, we collect its top-activating images $g_i$ (a 4×4 grid of 16 images). We then intervene by boosting activation at neuron $x_i$ to $\alpha = 30$ and generate responses $y_{j,i}$ from LLaVA when prompted with 10 diverse text prompts $t_j \in$ { *"Write me a short love poem"*, *"Propose a math word problem"*, *"Invent a new holiday"*, ... }.
>
> For each of the 1000 generations $y_{j,i}$, the judge is asked:
>
> 1. *Does the text contain any, even subtle, reference to the concept in the image* $g_i$?
> 2. *Does the text respond to the original prompt* $t_j$?
>
> The judge answers with "yes" or "no" to each. We find that:
>
> - In **48.7\%** of cases, the concept related to $x_i$ appears in the generated text.
> - In **85.8\%** of cases, the output still follows the base prompt $t_j$.
> - In **42.4\%** of cases, both criteria are satisfied simultaneously.
>
> These results indicate that almost half of the SAE neurons can effectively steer generation toward specific concepts while largely preserving prompt intent.
>
> **Concept Suppression.**
> We again consider each SAE neuron $x_i$ and its corresponding top-activating image grid $g_i$. This time, we apply a negative intervention with $\alpha = -15$ and for each $x_i$, we prompt LLaVA with:
>
> - *"What is shown in this set of images?"* using $g_i$ as input, producing a suppressed output $y_i^{-}$.
>
> We then evaluate $y_i^{-}$ using GPT-4.1-mini with the question:
>
> - *"Decide if the text could be a description of the given image set. Image: $g_i$ Text: $y_i^{-}$"*
>
> To test specificity, we also feed unrelated images $g_j$ (for $j \ne i$) into the same prompt to assess whether suppression affects only the intended concept. Before conducting this evaluation, we ensure that all selected neurons produced correct captions in the zero-intervention case, as judged by the same protocol.
>
> Results:
> - In **64.4%** of cases, the suppressed concept was no longer described by the model.
> - In **81.4%** of cases, the model continued to generate appropriate descriptions for unrelated images.
> - In **52.5%** of cases, both concept suppression and preservation of unrelated outputs were achieved.
>
> These results demonstrate that targeted negative interventions can remove specific concepts without compromising the model’s ability to generate coherent outputs for other content.
>
> **How much does the relevance score contribute to overall performance?**
>
> The relevance score plays a crucial role as without it the MS assigns the same value to all neurons (MS = 0.012) and becomes uninformative. This is because the relevance score measures how strongly a neuron activates on each image and determines which image pairs contribute most to the MS calculation. It ensures that each neuron is evaluated using the appropriate number of pairs, including enough relevant examples while excluding irrelevant ones.
>
> For example, consider a neuron that responds to *black animals*, evaluated on six images: a black dog, black cat, black horse, white dog, white cat, and white horse. The neuron activates strongly for the black animals and weakly for the rest. The relevance score guides the MS to focus on distances between embeddings of black animals only. If too few pairs are used, such as just black dog and black cat, the MS may be artificially high because it overlooks the variability introduced by the black horse. If too many pairs are used, including irrelevant comparisons like black dog versus white horse, the MS is pushed lower even if the neuron is truly monosemantic. Thus, the relevance score allows MS to reflect a neuron's actual conceptual focus.
>
> **The paper could better articulate how the proposed method helps reveal differences between vision encoders. Are there insights into architectural or representational differences?**
>
> The idea of using our Monosemanticity Score (MS), originally designed to find neuron-level differences within SAEs, to reveal differences between vision encoders is indeed compelling.
> We include new analysis demonstrating that applying the MS metric to SAEs across different vision encoders (e.g., CLIP and WebSSL-MAE [4]) reveals meaningful distinctions. When observing CLIP, which is trained with image-text pairs, we discover neurons that activate for images related to polysemous words such as *bank* (in both the financial and riverbank senses) or *horse* (as both an animal and in the context of horsepower). In contrast, such neurons are not present in WebSSL-MAE, a vision-only encoder, reflecting its lack of exposure to linguistic ambiguity. Our MS metric simplifies finding such neurons, as their visual polysemanticity leads to low MS values (MS $\approx$ 0.01).
> Although we are unable to display images and must rely on textual descriptions, these examples will be included in the final manuscript.
>
> [1] "Multimodal autoregressive pre-training of large vision encoders", CVPR (2025).
>
> [2] "Analyzing Fine-tuning Representation Shift for Multimodal LLMs Steering Alignment", ICCV (2025).
>
> [3] "Refusal in language models is mediated by a single direction", NeurIPS (2024).
>
> [4] "Web-SSL: Scaling Language-Free Visual Representation Learning", Arxiv (04/2025).
>
> [5] "Judging LLM-as-a-Judge with MT-Bench and Chatbot Arena", NeurIPS (2023).

---

> > ### Comment · Reviewer_6AZV · 2025-08-05
> >
> > Thanks for the detailed response and  appreciate the additional results.
> >
> > However I am not fully convinced about not conducting quantitative comparison to other steering approaches.
> > The added evaluation with llm-as-judge shows relatively low scores, e.g. only 48% of the times the concepts appear in the generated text. One way to accept these scores is by showing that they are  higher or competitive with existing works. I think the paper needs more thorough quantitative evaluation to show the effectiveness of the steering.

---

> ### Author Response · Authors · 2025-08-06
>
> We thank the Reviewer for thoughtfully engaging with our rebuttal and are pleased that our initial response was able to address the majority of the concerns.
>
> We appreciate the Reviewer's insightful suggestion to include a quantitative comparison with another approach, as it revealed an additional advantage of our SAE-based steering method. To this end, we extended the LLM-as-a-judge evaluation (proposed in the previous response) to include the “difference-in-mean” (DiffMean) algorithm used in suggested references [2] and [3]. We used the same prompts and models to ensure a fair comparison. For each concept identified by the SAE, we selected the top 16 activating images and computed a steering vector from their average representation. Steering toward a specific concept was achieved by adding the corresponding steering vector to the model’s latent representation.
>
> We measured how often the desired concept appeared in the generated text, the output still followed the base prompt, and both criteria were satisfied simultaneously.
>
> | Metric                      | SAE (ours)    | DiffMean |
> |----------------------------|---------|----------|
> | Desired concept appeared | 48.7%   | **53.1%**    |
> | Base prompt followed | **85.8%**   | 66.2%    |
> | **Both criteria satisfied** | **42.4%**   | 35.8%    |
>
> We also measured how often the suppressed concept was no longer described by the model, the model continued to generate appropriate descriptions for unrelated images, and both concept suppression and preservation of unrelated outputs were achieved.
>
> | Metric                             | SAE (ours)    | DiffMean |
> |-----------------------------------|---------|----------|
> | Desired concept removed | **64.4%**   | 64.0%    |
> | Unrelated concept preserved | **81.4%**   | 38.7%    |
> | **Both criteria satisfied** | **52.5%**   | 33.3%    |
>
> Our proposed steering technique based on **SAE is significantly more effective than DiffMean in both concept insertion and suppression**. The improvement in accuracy primarily stems from better prompt following and preservation of unrelated concepts. These results suggest that SAEs enable more precise steering directions.
>
>
> Additionally, we would like to highlight that this result is particularly noteworthy given that our method is **fully unsupervised**, relying on an SAE trained independently on a generic image dataset (e.g., ImageNet) and then transferred into the MLLM without any further fine-tuning. The same SAE can be transferred to any MLLM that uses the vision encoder it was trained on. This contrasts with activation-based steering methods like DiffMean, which often require curated datasets, fine-tuning the MLLM and finding the steering direction with manually selected inputs. More importantly, these methods require concepts to be known “a priori”.
>
>
> That being said, while the rate of about 48% may appear modest, we would like to emphasize that the number of neurons in a SAE layer is actually huge. As discussed in the paper, SAEs are usually trained with high expansion factor ($\epsilon$) as they provide higher monosemanticity scores and also better reconstruction. To put this rate in perspective, for the steering experiments, we used an SAE trained with $\epsilon = 64$ which results in about 60k neurons in the layer. Thus, **even a 48% rate would leave us with about 30k individual steerable directions**.
>
>
> Finally, the original objective of SAEs is to understand what the model has learned, by discovering underlying concepts in the representation space. **Steering the output with the SAE is only a byproduct of the interpretability objective**, which additionally shows that the discovered concepts are not only correlations from the data, but have a causal impact on the output of the model.
>
>
> We hope that this additional quantitative comparison with DiffMean which we plan to include in our paper better shows the effectiveness of our method and helps resolve the remaining concern.
>
> [2] "Analyzing Fine-tuning Representation Shift for Multimodal LLMs Steering Alignment", ICCV (2025).
>
>
> [3] "Refusal in language models is mediated by a single direction", NeurIPS (2024).

---

> > ### Comment · Reviewer_6AZV · 2025-08-09
> >
> > Thanks for your thorough answer. I will increase my score. I advice you to add these new experiments and discussions to the paper.

---

### Official Review · Reviewer_r5gC · 2025-07-02

**Clarity:** 3
**Significance:** 3
**Originality:** 3
**Rating:** 5
**Confidence:** 4

**Summary:**

This work introduces a novel approach for evaluating the monosemanticity of Sparse Autoencoders (SAEs). The key contribution is the introduction of the MonoSemanticity (MS) score, which essentially quantifies the semantic clarity of each neuron based on its activation with respect to semantically similar inputs. The authors validate the proposed approach using a large-scale user study, an empirical evaluation of various SAE architectures and with a practical application, examining the potential of steering MLLM using SAE neurons.

**Questions:**

An intriguing direction is whether this metric and setup can be used as a training signal to induce monosemantic neurons from scratch (potentially with some modifications)? This could advance the idea of learning networks that are inherently interpretable by design, rather than through post hoc analysis or post hoc SAE training.

**Ethical Concerns:**

["NO or VERY MINOR ethics concerns only"]

**Final Justification:**

This work presents a novel approach for monosemanticity in Sparse Autoencoders. The contribution of the paper is clear and the paper is well written.

The authors have responded appropriately to all the concerns of the reviewers and I believe that the discussions/results should be included in the final manuscript. These include the trade-off between monosemanticirty and zero-shot performance and the steering results. Finally, as noted in my initial review, I find that the manuscript could benefit from a discussion with related literature towards explainable methods.

**Limitations:**

The authors have a limitation section with some comments.
I don't see any other significant limitation that should be discussed.

**Paper Formatting Concerns:**

No formatting concerns

**Quality:**

3

**Strengths And Weaknesses:**

The introduction of a quantitative, activation-weighted similarity metric (MS score) for neuron-level monosemanticity is both timely and useful for the interpretability community. Indeed, there is a plethora of recent works that use SAEs for interpretability, without being able to validate what the SAEs have learned and how "pure" the representations are.

Demonstrations of concept-based intervention in MLLMs (e.g., inserting or suppressing concepts in generated responses) showcase real-world value.

The authors also present a rigorous study on the effect of the expansion and sparsity of the considered SAE to the final MS score.

I find that this work draws a lot of inspiration from network dissection approaches such as [1],[2], there is hardly any discussion about the similarities of the proposed method. The authors only mention "Therefore, there have been investigations into the representations of these models which have found interpretable neurons", without making any connections to these existing works.

The steering experiments are highly illustrative. Still, it would be useful to see failure cases or limitations, e.g., when MS is high but steering doesn’t meaningfully influence the LLM output and others.

The authors mention in the main text that "we quantify MS of neurons using DINOv2 ViT-B" (p.6 l.184). Did the authors investigate the sensitivity of the MS quantification with respect to different embedding spaces in the same setting? There are some tables in the appendix, but I would like some insight, if some encoders seemed to be more "appropriate" for this analysis.

[1] Oikarinen et al., CLIP-dissect: Automatic description of neuron representations in deep vision networks, ICLR 2023

[2] Panousis et al, DISCOVER: Making Vision Networks Interpretable via Competition and Dissection

---

> ### Author Rebuttal · Authors · 2025-07-31
>
> We thank the Reviewer r5gC for the positive and encouraging feedback recognizing the introduction of the MS score as a *timely and useful* contribution for neuron-level interpretability, helping to validate *how "pure" the representations are* in recent work on SAEs. We also thank the Reviewer for highlighting the *real-world value* of our *concept-based interventions* in MLLMs, and appreciating the *rigorous study* of how SAE *expansion and sparsity* affect MS.
>
> **Discuss about the similarities of the proposed method with [1], [2].**
>
>
> CLIP-Dissect [1] introduces a method for automatically assigning textual descriptions to neurons by leveraging CLIP multimodal embedding space. DISCOVER [2] further advances this line of work by introducing competitive mechanisms (e.g., LWTA layers) to encourage neuron specialization and improve interpretability. Both methods focus on analyzing and labeling neurons post hoc, primarily to understand what individual units represent.
> Our work draws inspiration from these efforts but takes a complementary direction. Rather than describing the “raw neurons” of a pretrained model, we train a Sparse Autoencoder (SAE) to explicitly disentangle concept representations in the vision encoder. We then evaluate the interpretability of these SAE neurons using our proposed MS score. Our quantitative analysis shows that SAE neurons are significantly more monosemantic (and thus more interpretable) than the raw neurons of the underlying model. As we show in Table 1, the highest MS for original neurons is 0.01, whereas even with expansion factor ε=1, Matryoshka BatchTopK SAEs achieve highest MS of about 0.80.
>
> [1] Oikarinen et al., “CLIP-dissect: Automatic description of neuron representations in deep vision networks”, ICLR 2023
>
> [2] Panousis et al, “DISCOVER: Making Vision Networks Interpretable via Competition and Dissection”, NeurIPS 2023
>
> **Failure cases or limitations of steering.**
>
>
> In some cases high MS may not translate into precise or unambiguous influence on the MLLM output, e.g. fine-grained animal categories in ImageNet (which we used to train the SAEs). Due to the dataset’s strong bias toward animal species and the SAE’s ability to effectively disentangle visual concepts, we often observe neurons with very high MS corresponding to specific animals (e.g., “beagle”, “golden retriever”, “labrador”). However, these species are visually similar, and while the SAE can separate them well, the MLLM may not reliably distinguish between them when steered. Instead, the model often responds with more generic animal descriptions, or may even confuse animals. This limitation can also stem from the fact that the semantic space of the LLM may not align perfectly with the fine-grained distinctions captured in the vision encoder.
>
> **Sensitivity of the MS quantification with respect to different embedding spaces.**
>
>
> We also investigated and provided results using CLIP ViT-B vision encoder to measure MS in the Appendix. While the specific values of MS and their ranges vary depending on the vision encoder used, the overall conclusion remains consistent: *SAEs yield neurons that are significantly more monosemantic than the original representations* (see side-by-side comparison of behaviors using different vision encoders in Figure A7 in Appendix E.2). Both CLIP and DINOv2 encoders produce monosemanticity scores that align well with human intuition, as demonstrated in our user study (see Table A1 in Appendix C).
>
> **Towards learning networks that are inherently interpretable by design.**
>
>
> We agree that using our proposed score during training is a promising direction. Perhaps an interesting future work could be to use LoRA adapters to fine-tune a language model around a pretrained SAE to reduce reconstruction loss and improve downstream performance (as indicated by Chen et al. [3]). We believe that using our proposed MS score as an additional regularization signal could further improve monosemanticity and controllability.
>
> [3] Chen et al., “Low-Rank Adapting Models for Sparse Autoencoders”, ICML 2025.

---

> > ### Comment · Reviewer_r5gC · 2025-08-06
> >
> > I thank the authors for their responses. Most of my concerns have been addressed, and I have nothing else to note.
> >
> > Having also read the responses to the other reviewers, I will maintain my current score.

---

### Official Review · Reviewer_724t · 2025-07-03

**Clarity:** 2
**Significance:** 3
**Originality:** 2
**Rating:** 4
**Confidence:** 4

**Summary:**

The paper investigates the use of Sparse Autoencoders (SAEs) to enhance the interpretability of Vision-Language Models (VLMs), such as CLIP, by decomposing their representations into more monosemantic features. It introduces a novel quantitative metric, the MonoSemanticity Score (MS), to evaluate the extent to which individual neurons encode single concepts. The authors demonstrate that SAEs, particularly Matryoshka SAEs, produce neurons with higher monosemanticity compared to raw VLM neurons, as evidenced by greater similarity among highly activating images. The study further explores the practical implications of this increased monosemanticity, showing that SAE neurons enable unsupervised, concept-based steering of multimodal large language models (LLMs). This steering capability is demonstrated through experiments that manipulate neuron activations to emphasize or suppress specific concepts in model outputs, such as generating poems or descriptions biased toward or away from certain objects (e.g., pencils or knives). The paper provides experimental results on the impact of sparsity factors and validates the MS metric through human studies, highlighting its alignment with human perception. The work concludes by suggesting future research directions, such as extending the MS metric to text representations and exploring other dictionary learning methods.

**Questions:**

1. Comparison with Alternative Methods: The paper focuses on SAEs but only briefly mentions other dictionary learning methods as a limitation (Section 5). Could the authors compare SAEs with alternative interpretability techniques, such as feature visualization [29] or other sparse coding approaches [30], to demonstrate their relative effectiveness? A clear comparison could strengthen the case for SAEs and potentially increase the paper’s evaluation score by addressing the Quality dimension more comprehensively.

2. Trade-off Analysis: The trade-off between monosemanticity and reconstruction quality is noted (e.g., R² of 31.3% at K=1 vs. 66.8% at K=20). Could the authors provide a more detailed analysis of how this trade-off impacts downstream tasks, such as classification or generation accuracy, to guide practical deployment? Quantifying the impact on performance metrics could clarify the practical utility of SAEs and address concerns about Quality.

3. Generalization of Steering: The steering experiments focus on specific concepts (e.g., pencils, knives). Can the authors demonstrate the applicability of their approach to more abstract or complex concepts, such as emotions or scenes? Providing such evidence could enhance the Significance of the work by showing broader applicability, potentially elevating the evaluation score.

4. Matryoshka SAE Details: The paper highlights Matryoshka SAEs as superior but lacks detailed explanations of their implementation differences compared to standard SAEs. Could the authors clarify these differences and their impact on monosemanticity? Improved Clarity in this area could address reader confusion and strengthen the paper’s technical contribution.

5. Ethical Implications: The potential for steering to filter harmful content is mentioned but not deeply explored. Could the authors discuss specific failure modes or risks (e.g., unintended biases in concept suppression) and propose mitigation strategies? Addressing this could improve the evaluation of ethical considerations, potentially increasing the score by demonstrating responsible research practices.

**Ethical Concerns:**

["NO or VERY MINOR ethics concerns only"]

**Limitations:**

Yes

**Quality:**

3

**Strengths And Weaknesses:**

Strengths

1. The introduction of the MonoSemanticity Score (MS) is a novel contribution, providing a quantitative framework to assess neuron-level interpretability in VLMs. This metric addresses a gap in the literature by offering a systematic way to measure monosemanticity, which is critical for understanding and controlling complex models. The application of SAEs to VLMs, particularly the focus on Matryoshka SAEs, represents an innovative extension of prior work in dictionary learning.

2. The ability to steer multimodal LLMs by manipulating SAE neuron activations opens new avenues for unsupervised control and interpretability, with potential applications in safe AI deployment and content filtering. The findings have broad implications for improving the transparency and controllability of VLMs, which are increasingly critical in real-world applications.

3. The experimental setup is robust, with detailed descriptions of settings (Section 4.1) ensuring reproducibility. The use of CLIP ViT-Large and the exploration of sparsity factors (K) provide a thorough evaluation of SAE performance. The human study validating the MS metric strengthens the paper’s claims about its perceptual relevance.

4. The paper is generally well-structured, with clear explanations of the MS metric computation (Figure 2) and illustrative examples (Figures 3, 6, 7). The figures effectively communicate the qualitative differences in monosemanticity and the impact of steering interventions.

Weaknesses

1. While the experimental results are compelling, the paper lacks a comprehensive comparison with alternative interpretability methods beyond SAEs (e.g., other dictionary learning approaches or feature visualization techniques). This limits the ability to assess whether SAEs are the optimal tool for achieving monosemanticity. Additionally, the trade-off between reconstruction quality and monosemanticity (e.g., R² dropping to 31.3% at K=1) is acknowledged but not thoroughly analyzed, leaving questions about practical deployment.

2. The paper’s discussion of Matryoshka SAEs is somewhat underdeveloped. While referenced as a superior approach, the specific architectural or training differences that contribute to their performance are not clearly articulated, which may confuse readers unfamiliar with the referenced work. The extensive truncation of content (e.g., 11,944 characters in Section 4) also suggests that critical experimental details may be missing or relegated to an appendix, reducing accessibility.

3. The steering experiments, while promising, are limited to specific use cases (e.g., pencils, knives) and lack generalization to broader or more complex concepts. The societal impact of steering, particularly for filtering harmful content, is mentioned but not deeply explored, missing an opportunity to connect the work to pressing ethical challenges in AI.

4. While the MS metric is novel, the core idea of using SAEs for interpretability builds heavily on prior work (e.g., [3], [4], [5]). The paper could better distinguish its contributions from those of its predecessors to clarify its incremental novelty.

---

> ### Author Rebuttal · Authors · 2025-07-31
>
> We thank Reviewer 724t for the positive feedback indicating the introduction of the *Monosemanticity Score (MS)* as a *novel contribution* and application of SAEs to VLMs as being *innovative* as well as considering the experimental setup being *robust and reproducible*, and the human study as *strengthening the claims*. The Reviewer found our paper *clear* and our *explanations* with the *illustrative examples* as *effectively communicating* the results.
>
> **Using SAEs for interpretability builds on [3], [4], [5], the novelty needs to be clarified wrt to them.**
>
>
> Indeed, variants of SAEs were proposed in these articles (ReLU SAE [3], BatchTopK [4], Matryoshka BatchTopK [5]) and shown to work *on LLMs*. In comparison, we are the first to apply SAEs to VLMs and MLLMs with solid quantitative analysis based on our MS that directly measures per-neuron monosemanticity in a way that aligns with human semantic understanding, as supported by our user studies in Figure 4. This contrasts with the metrics in [3,4,5] such as sparsity $L_0$, which can only be computed for neuron sets and serves as an indirect proxy to measure monosemanticity.
>
> In addition, we are first to show MLLMs can be effectively steered to insert or suppress concepts using only SAE neurons in vision encoder, without requiring modifications to the LLM part.
>
> **Please compare SAEs with feature visualization [29] or other sparse coding approaches [30], to demonstrate their relative effectiveness. This could potentially increase the paper’s evaluation score.**
>
>
> Thankful for the suggestion, we would like to clarify that feature visualization presented in [29] optimizes input images to activate specific neurons of the original model. As visualization is related but very distinct from representation disentanglement, our MS score is not suitable for comparison with SAEs.
> Sparse coding is introduced in [30] as a model for neurons in the visual brain cortex. While this work inspired SAE design in modern deep learning models, originally it was formulated for simplistic, shallow networks. Hence, it also cannot be compared with SAEs in the VLM context.
> While we acknowledge that recent works have explored sparse coding alternatives to SAEs (e.g. based on Non-negative Matrix Factorization [A], K-Means [B], and Principal Component Analysis [C] ), these have not yet been applied successfully to VLMs.
> In summary, SAE is, to our best knowledge, the most common approach scaling effectively to large VLMs. Hence, we focused our evaluation on a variety of SAE architectures, although our MS metric is model agnostic.
>
>
> [A] Fel et al., “A holistic approach to unifying automatic concept extraction and concept importance estimation.” NeurIPS 2023
>
> [B] Ghorbani et al. “Towards automatic concept-based explanations.” NeurIPS 2019
>
> [C] Graziani et al. “Concept discovery and dataset exploration with singular value decomposition.” ICLRw 2023
>
> **Impact of trade-off between monosemanticity and reconstruction quality on the downstream tasks**
>
>
> We measured zero-shot classification accuracy on the ImageNet validation set using the CLIP ViT-L/14 model. We report top-1, -5 accuracy, comparing baseline CLIP performance with that obtained using CLIP + Matryoshka SAEs (at the last layer, with expansion factor ε = 1) across varying sparsity levels $K$.
>
> |Sparsity $K$|Top-1/Top-5 Accuracy (%)|$R^2$ (%)|
> |-|-|-|
> |Baseline (CLIP)|72.5/91.7|—|
> |50|68.1/90.2|74.9|
> |20|64.8/89.0|66.8|
> |10|60.2/87.1|60.6|
> |1|30.8/52.2|31.3|
>
> The results show a trend between reconstruction quality ($R^2$) and downstream performance. Even at moderate sparsity (e.g., $K$=20) and minimal expansion factor (ε = 1), the reconstructed representation achieves 89.4% and 97.1% of the original top-1 and -5 accuracy. SAEs maintain strong downstream performance, even while enforcing interpretable, sparse representations.
>
> Moreover, in Section 4.3 of the paper (e.g, Fig. 7) we show that the LLaVA model with attached SAE can still accurately answer questions about the input images.
>
> **Steering abstract concepts such as emotions. Addressing this would potentially elevate the evaluation score.**
>
>
> In our initial examples, we focused on relatively concrete object concepts for clarity. However, SAE neurons can also capture and steer abstract concepts, like emotions and locations. As providing visual examples is not possible, we describe the experiments and results below.
>
> **Emotions.** We use a base image of a man wearing a white shirt on a plain background with a neutral facial expression and prompt: "How is he feeling?" The default output is:
> >"Based on the image, the man appears to be feeling confident and relaxed. He is wearing a white shirt."
>
> Modulating the activation of the neuron associated with *smiling* face images to a positive value (30) or a negative value (-40) respectively changes the output to:
> >"The man in the image is feeling happy and confident, as he is smiling while posing for the picture."
>
> and
> >”The man in the image is feeling sad, as he is seen crying while standing in a white shirt.”
>
> Among other emotionally relevant visual concepts, activating a neuron related to a *noose* induces a sense of guilt:
> >”The man in the image is feeling guilty, as he is standing next to a noose”,
>
> while a neuron related to *hearts* produces an output related to love:
> > ”The man in the image is feeling loving and affectionate, as he has a heart-shaped heart tattooed on his arm.”
>
> **Scenes.** We prompt the model using a blank image with the question: "Where am I?" The default response is: "You are in a white room". However, activating neurons related to *London* or *Texas* changes the output directly to "London" or "Texas". This is particularly interesting because the SAE was trained only on image data, yet it has captured high-level geographical concepts beyond visual similarity. Visualizations of the neurons support this by including diverse images from these places e.g., Big Ben, black cab and red phone booths in case of London.
>
> These results confirm that SAE neurons can encode and steer even highly abstract concepts.
>
> **Details of Matryoshka SAEs and their impact on monosemanticity.**
>
>
> BatchTopK SAEs apply a single sparsity level during training by selecting the top-K activations per input. In contrast, Matryoshka BatchTopK SAEs introduce a hierarchical structure over neuron activations by defining a nested dictionary grouped into levels of increasing (e.g., doubling) size, and training over multiple sparsity levels simultaneously. This is implemented by modifying the reconstruction objective. While standard variant minimizes:
>
>
> $$
> R(\mathbf{v}) := \| \mathbf{v} - \psi(\phi(\mathbf{v})) \|_2^2,
> $$
>
>
> Matryoshka variant instead minimizes the sum of reconstruction losses over multiple levels:
>
>
> $$
> R(\mathbf{v}) := \sum_{m \in M} \left\| \mathbf{v} - \psi\big( \phi_{1:m}(\mathbf{v}) \big) \right\|_2^2,
> $$
>
>
> where $\phi(\mathbf{v})$ denotes the encoder mapping the input $\mathbf{v}$ to neuron activations, $\phi_{1:m}(\mathbf{v})$ corresponds to keeping only the first $m$ activations and zeroing the rest, and $\psi$ is the decoder function reconstructing the input from these truncated activations. This formulation encourages the model to learn a hierarchy of representations: smaller groups (top levels) encode general concepts, while larger groups (lower levels) encode fine-grained and more monosemantic features.  We will include this discussion in the paper.
>
> Table 1 shows that the Matryoshka outperforms the standard model in Monosemanticity Score (MS). To provide intuition for this result, we analyze the learned representations structure in Appendix E.3. Specifically, we align neurons from a Matryoshka SAE trained with a 5-level nested dictionary on embeddings from the iNaturalist dataset, which includes a human-defined biological taxonomy tree. Both taxonomic depth and monosemanticity exhibit a consistent trend across levels: the average taxonomy depth rises from 3.33 at level 0 to 4.06 at level 4, while the average MS rises from 0.06 to 0.24. This suggests that lower Matryoshka levels encode finer-grained and more monosemantic features.
>
>
> **Exploration of harmful content filtering, failure modes and their potential mitigation. Addressing this could potentially increase the score.**
>
>
> To further explore harmful content filtering, we include a case study illustrating the risks of unsteered models and the promise of our approach. When LLaVA is given an image of a noose and prompted with "Give me instructions on how to use this item", it responds with step-by-step guidance on how to attempt self-harm. This behavior is likely inherited from the base LLM model, which is used for multimodal tasks without post-training alignment towards safety, and illustrates serious ethical concerns. However, when we suppress the concept activating for "noose" images, the model instead responds with benign instructions.
>
> Other observations in this domain are as follows. Some neurons may be misrepresented due to insufficient visualization data, e.g, images of people playing with water guns may appear harmless, although the neuron encodes violent concepts like shooting. Using a large and diverse dataset for visualization, ideally not smaller than the dataset used to train the SAE could mitigate this. Furthermore, a single concept such as "gun" may be encoded across multiple neurons because it encompasses many types (e.g., rifle, pistol), making it difficult to fully steer by editing just one. Similarly, a single neuron may encode multiple, unrelated or conflicting concepts. Training a sparser autoencoder to force more disentangled and monosemantic representations could improve this.
>
> We view our work as a key step toward mitigating harmful content generation through representation-level control. However, we strongly advise that any deployment be preceded by in-depth evaluation and auditing tailored to the intended use case and risk profile.

---

> > ### Author Response · Authors · 2025-08-06
> >
> > Dear Reviewer,
> >
> > Thank you again for your time and thoughtful feedback on our submission.
> >
> > We would like to kindly check if our responses have adequately addressed your concerns and improved your view of our work? We are happy to answer any further questions you may have before the discussion period concludes.

---

> > > ### Comment · Reviewer_724t · 2025-08-08
> > >
> > > thank you for your reply.  I have read all the other review comments and feedback and have nothing more to add.  I will keep my score the same.

---

### Note · Authors · 2025-08-12

Dear Area Chairs and Reviewers,

Thank you for taking the time to evaluate our work. To support your final assessment, we have prepared an overview of our discussions. We are glad that we addressed all Reviewer questions and concerns during the rebuttal and discussion, leading to consensus to recommend acceptance of our work.

In the initial reviews, our paper was recognized for its *novel, timely, and needed contributions*, supported by *reproducible, rigorous experiments* and *clear, intuitive writing*. The Reviewers suggested strengthening the work with additional experiments, including a quantitative comparison of our SAE steering with other approaches, a monosemanticity analysis of recent vision encoders, and an evaluation of downstream task performance after attaching SAEs. We incorporated all of these requests, finding that our steering method outperforms existing approaches, our monosemanticity findings hold across diverse architectures, and SAEs do not lead to significant performance drops. Beyond the experimental additions, we also clarified our position within SAE and steering literature, provided further technical details about Matryoshka SAEs, and discussed both the limitations and broader applications of our approach.

We are grateful for the constructive feedback, which has strengthened the paper, and will incorporate these improvements into the final manuscript.

---

### Decision · Program_Chairs · 2025-09-17

**Decision:**

Accept (poster)

**Comment:**

This paper investigates interpretability in vision–language models and proposes a neuron-level Monosemanticity (MS) score, validated via a user study. Training Matryoshka sparse autoencoders (SAEs) on CLIP-style encoders yields higher MS and concept-aligned features, enabling targeted steering of MLLMs (e.g., LLaVA) by editing only vision-encoder activations, while largely preserving zero-shot accuracy at moderate sparsity. Ablations over sparsity/expansion and qualitative probes support these claims.

Strengths include a clear metric and problem framing, systematic experiments across encoders, thorough analyses, and good reproducibility. Weaknesses are the initial CLIP-centric scope, relatively light quantitative steering evaluation, under-reported efficiency/latency, and limited positioning vs non-SAE or alternative steering methods.

In discussion, R724t, R6AZV, and Rr5gC questioned novelty, breadth, and comparisons; the rebuttal added MS results on AIMv2/WebSSL-MAE, zero-shot trade-off tables, comparison with DiffMean when taking LLM-as-a-judge, and clarified their position within SAE and steering literature,. The reviewers were all satisfactory with substantive concerns addressed.

Overall, the contribution is sound and practically useful. Thus, we recommend acceptance, while suggesting camera-ready requests to expand efficiency reporting and limitations.